# Tyr1 phosphorylation promotes phosphorylation of Ser2 on the C-terminal domain of eukaryotic RNA polymerase II by P-TEFb

Joshua E Mayfield[1†‡], Seema Irani[2†], Edwin E Escobar[3], Zhao Zhang[4], Nathaniel T Burkholder[1], Michelle R Robinson[3], M Rachel Mehaffey[3], Sarah N Sipe[3], Wanjie Yang[1], Nicholas A Prescott[1], Karan R Kathuria[1], Zhijie Liu[4], Jennifer S Brodbelt[3], Yan Zhang[1,5]*

[1]Department of Molecular Biosciences, University of Texas at Austin, Austin, United States; [2]Department of Chemical Engineering, University of Texas at Austin, Austin, United States; [3]Department of Chemistry, University of Texas at Austin, Austin, United States; [4]Department of Molecular Medicine, Institute of Biotechnology, University of Texas Health Science Center at San Antonio, San Antonio, United States; [5]Institute for Cellular and Molecular Biology, University of Texas at Austin, Austin, United States

*For correspondence:
jzhang@cm.utexas.edu

[†]These authors contributed equally to this work

Present address: [‡]Department of Pharmacology, University of California, San Diego, San Diego, United States

Competing interests: The authors declare that no competing interests exist.

**Abstract** The Positive Transcription Elongation Factor b (P-TEFb) phosphorylates Ser2 residues of the C-terminal domain (CTD) of the largest subunit (RPB1) of RNA polymerase II and is essential for the transition from transcription initiation to elongation in vivo. Surprisingly, P-TEFb exhibits Ser5 phosphorylation activity in vitro. The mechanism garnering Ser2 specificity to P-TEFb remains elusive and hinders understanding of the transition from transcription initiation to elongation. Through in vitro reconstruction of CTD phosphorylation, mass spectrometry analysis, and chromatin immunoprecipitation sequencing (ChIP-seq) analysis, we uncover a mechanism by which Tyr1 phosphorylation directs the kinase activity of P-TEFb and alters its specificity from Ser5 to Ser2. The loss of Tyr1 phosphorylation causes an accumulation of RNA polymerase II in the promoter region as detected by ChIP-seq. We demonstrate the ability of Tyr1 phosphorylation to generate a heterogeneous CTD modification landscape that expands the CTD's coding potential. These findings provide direct experimental evidence for a combinatorial CTD phosphorylation code wherein previously installed modifications direct the identity and abundance of subsequent coding events by influencing the behavior of downstream enzymes.
DOI: https://doi.org/10.7554/eLife.48725.001

## Introduction

The C-terminal domain of the RPB1 subunit of RNA polymerase II (CTD) is composed of a species-specific number of repeats of the consensus amino acid heptad YSPTSPS (arbitrarily numbered as Tyr1, Ser2, Pro3, Thr4, Ser5, Pro6, and Ser7) (*Jeronimo et al., 2016*). The CTD undergoes extensive post-translational modification (PTM) that recruits RNA processing and transcription factors that regulate progression through the various stages of transcription. These modification events are dynamic, highly regulated, and maintained through the complex interplay of CTD modification enzymes. Collectively these PTMs and recruited protein factors constitute the 'CTD Code' for eukaryotic transcription (*Buratowski, 2003*).

**eLife digest** DNA contains the instructions for making proteins, which build and maintain our cells. So that the information encoded in DNA can be used, a molecular machine called RNA polymerase II makes copies of specific genes. These copies, in the form of a molecule called RNA, convey the instructions for making proteins to the rest of the cell.

To ensure that RNA polymerase II copies the correct genes at the correct time, a group of regulatory proteins are needed to control its activity. Many of these proteins interact with RNA polymerase II at a region known as the C-terminal domain, or CTD for short. For example, before RNA polymerase can make a full copy of a gene, a small molecule called a phosphate group must first be added to CTD at specific units known as Ser2.

The regulatory protein P-TEFb was thought to be responsible for phosphorylating Ser2. However, it was previously not known how P-TEFb added this phosphate group, and why it did not also add phosphate groups to other positions in the CTD domain that are structurally similar to Ser2.

To investigate this, Mayfield, Irani et al. mixed the CTD domain with different regulatory proteins, and used various biochemical approaches to examine which specific positions of the domain had phosphate groups attached. These experiments revealed a previously unknown aspect of P-TEFb activity: its specificity for Ser2 increased dramatically if a different regulatory protein first added a phosphate group to a nearby location in CTD. This additional phosphate group directed P-TEFb to then add its phosphate specifically at Ser2.

To confirm the activity of this mechanism in living human cells, Mayfield, Irani et al. used a drug that prevented the first phosphate from being added. In the drug treated cells, RNA polymerase II was found more frequently 'stalled' at positions on the DNA just before a gene starts. This suggests that living cells needs this two-phosphate code system in order for RNA polymerase II to progress and make copies of specific genes.

These results are a step forward in understanding the complex control mechanisms cells use to make proteins from their DNA. Moreover, the model presented here – one phosphate addition priming a second specific phosphate addition – provides a template that may underlie similar regulatory processes.

DOI: https://doi.org/10.7554/eLife.48725.002

Chromatin immunoprecipitation and next-generation sequencing technologies have revealed how phosphorylation levels of CTD residues change temporally and spatially during each transcription cycle (*Eick and Geyer, 2013*). The major sites of phosphorylation are Ser5 and Ser2, directed by Transcription Factor II H (TFIIH) (*Feaver et al., 1994*) and P-TEFb in mammals (*Marshall et al., 1996*), respectively. The other three phosphate-accepting residues (Tyr1, Thr4, and Ser7) are also subject to modification, although their functions are less well-understood (*Jeronimo et al., 2013*). In mammalian cells, the phosphorylations of Tyr1 and Ser7 rise and peak near the promoter along with Ser5 and gradually decrease as transcription progresses towards termination. The phosphorylation of Thr4 and Ser2, on the other hand, don't appear until later in the transcription cycle during elongation (*Eick and Geyer, 2013*). The molecular underpinnings resulting in this orchestration are poorly defined. A particularly apparent gap in current knowledge is if sequence divergence from the consensus heptad or previously installed PTMs influence coding events.

The CTD code is generated through the interplay of CTD modifying enzymes such as kinases, phosphatases, and prolyl isomerases (*Bataille et al., 2012*). Disruption of this process is implicated in various disease states. P-TEFb is of particular interest due to its overexpression in multiple tumor types and role in HIV infection (*Franco et al., 2018*). As a major CTD kinase, P-TEFb promotes transcription by contributing to the release of RNA polymerase II from the promoter-proximal pause through its phosphorylation of Negative Elongation Factor (NELF), DRB Sensitivity Inducing Factor (DSIF), and Ser2 of the CTD (*Wada et al., 1998*). Interestingly, P-TEFb seems to phosphorylate Ser5 of the CTD in vitro and mutation of Ser5 to alanine prevents the phosphorylation of CTD substrates. However, mutation of Ser2 to alanine did not result in this abolishment (*Czudnochowski et al., 2012*). These results are in contrast to in vivo studies of P-TEFb specificity, where compromised P-TEFb kinase activity results in a specific reduction in levels of Ser2 phosphorylation

(*Marshall et al., 1996*). The discrepancies between P-TEFb specificity in vitro and in vivo make it difficult to reconcile P-TEFb's function as a CTD Ser2 kinases (*Bartkowiak et al., 2010*; *Czudnochowski et al., 2012*).

To resolve these inconsistencies, we utilize a multi-disciplinary approach to investigate the specificity of P-TEFb. Identification of phosphorylation sites was carried out using ultraviolet photodissociation (UVPD) mass spectrometry establishing the specificity of P-TEFb in vitro in single residue resolution. We reveal the tyrosine kinase c-Abl phosphorylates consensus and full-length CTD substrates in a conservative fashion, with only half of the available sites phosphorylated. The unique phosphorylation pattern of Tyr1 by tyrosine kinases like c-Abl directs the specificity of P-TEFb to Ser2. The priming effect of pTyr1 on P-TEFb extends to human cells, where small-molecule inhibition of c-Abl-like Tyr1 kinase activities leads to a reduction of Tyr1 phosphorylation. Further ChIP-seq analysis shows that the loss of tyrosine phosphorylation increases promoter-proximal pausing with an accumulation of RNA polymerase II at the promoter region of the gene. Overall, our results reconcile the discrepancy of P-TEFb kinase activity in vitro and in cells, showing that Tyr1 phosphorylation can prime P-TEFb and alter its specificity to Ser2. Importantly, these findings provide direct experimental evidence for a combinatorial CTD phosphorylation code wherein previously installed modifications direct the identity and abundance of subsequent coding events, resulting in a varied PTM landscape along the CTD allowing for diversified co-transcriptional signaling.

## Results

### Determination of P-TEFb specificity in vitro using mass spectrometry

To define P-TEFb's specificity directly on full-length RPB1 CTD substrates, we applied matrix-assisted laser desorption/ionization-mass spectrometry (MALDI-MS) and liquid chromatography ultraviolet photodissociation tandem mass spectrometry (LC-UVPD-MS/MS) to identify the substrate residues of this kinase. Ultraviolet photodissociation (UVPD) using 193 nm photons is an alternative to existing collision- and electron-based activation methods in proteomic mass spectrometry. This method energizes peptide ions via a single absorption event of high-energy photons resulting in a greater number of diagnostic fragment ions and the conservation of lower energy bonds like those of some PTMs including phosphorylations (*Brodbelt, 2014*). This method is applicable in both positive and negative ionization modes, results in a greater degree of peptide fragmentation, better certainty in PTM localization, and conservation of PTMs to ultimately ensure the detection of even low abundance or particularly labile modifications. Because endogenous RNA polymerase II is heterogeneously modified, we used recombinant yeast CTD GST fusion proteins, which contain mostly consensus heptad repeats (20 of 26), as an unmodified substrate for PTM analysis (*Figure 1—figure supplement 1*). The stability and consistency of GST yeast CTD (yCTD) make it ideal for studying CTD modification along consensus heptads. With high kinase and ATP concentration (2 mM) and overnight incubation (~16 hr), P-TEFb generates two phospho-peptides as detected by LC-UVPD-MS/MS: a major species phosphorylated on Ser5 ($Y_1S_2P_3T_4pS_5P_6S_7$) and a minor species phosphorylated on Ser2 ($S_5P_6S_7Y_1pS_2P_3T_4$) (*Figure 1A* and *Figure 1—figure supplement 2A–B*). This is highly similar to patterns observed previously for *bona fide* Ser5 CTD kinases Erk2 and TFIIH (*Mayfield et al., 2017*). These experiments confirm P-TEFb's inherent in vitro preference for Ser5 when phosphorylating unmodified CTD (*Czudnochowski et al., 2012*; *Portz et al., 2017*).

We next measured the total number of phosphates added to the CTD by P-TEFb. MALDI-MS analysis of yCTD treated with P-TEFb reveals a cluster of peaks with mass shifts relative to no kinase control ranging from 1906.1 to 2318.4 Da, each interspaced by 80 Da (*Figure 1B* and *Figure 1—figure supplement 2C*). This corresponds to the addition of 24 to 29 phosphates to yCTD's 26 heptad repeats. This finding in combination with our LC-UVPD-MS/MS analysis of P-TEFb treated yCTD indicates that P-TEFb phosphorylates the CTD in an average one phosphorylation per heptad manner, and these heptads are primarily phosphoryl-Ser5 (pSer5) in vitro.

### Amino acid identity at the Tyr1 position is important for kinase activity

To phosphorylate Ser2 and Ser5, CTD kinases must discriminate very similar SP motifs in the CTD, $Y_1\underline{S_2}P_3$ and $T_4\underline{S_5}P_6$, to maintain accuracy during transcription. Among the flanking residues of these two motifs, the unique structure of the tyrosine side chain likely contributes to the recognition of the

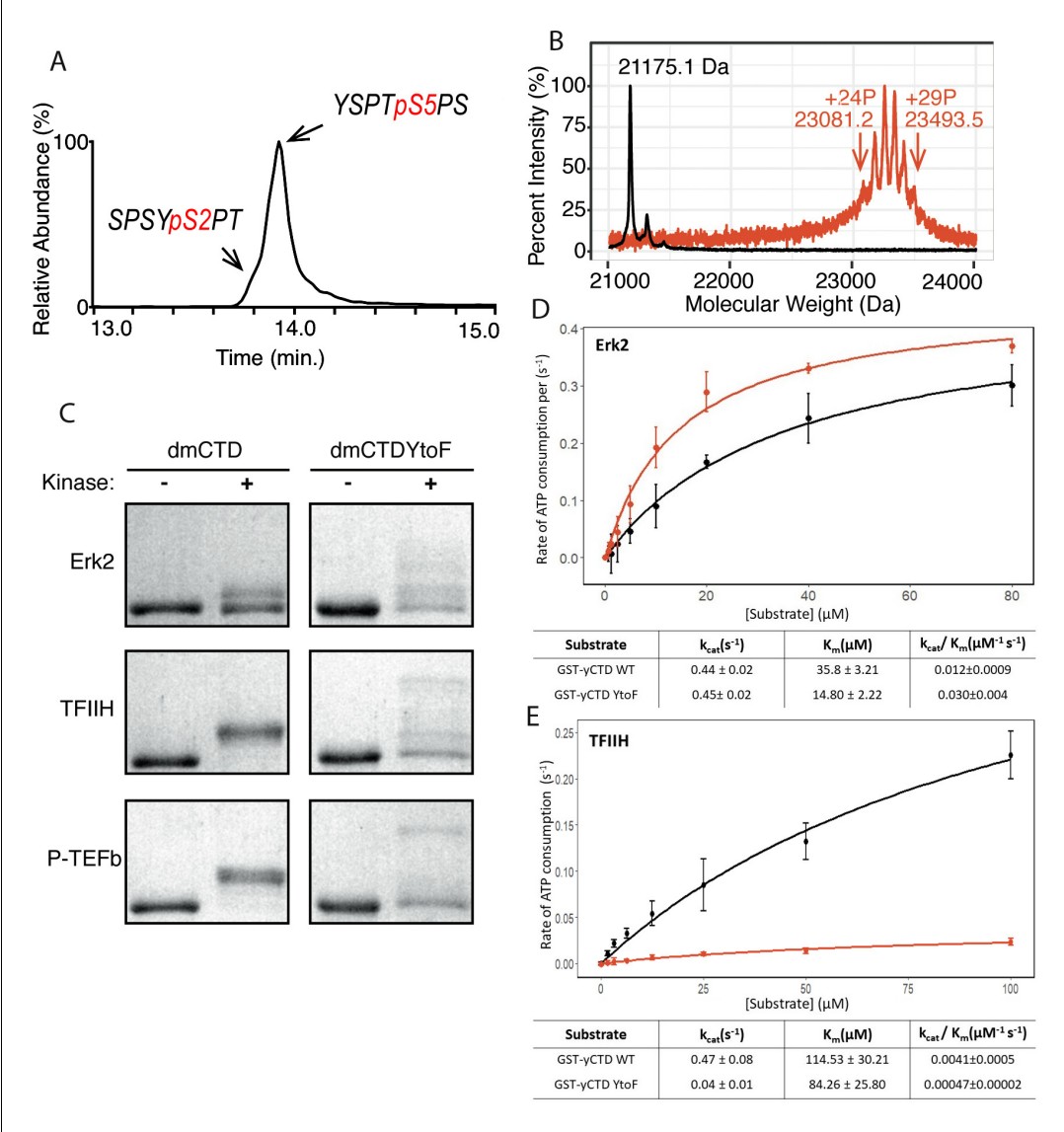

**Figure 1.** P-TEFb in vitro activity and the effect of phenylalanine replacement of Tyr1 in CTD phosphorylation. (A) LC-UVPD-MS/MS analysis of yCTD treated with P-TEFb alone showing extracted ion chromatogram for two CTD heptads. (B) Portions of MALDI mass spectra of 3C-protease digested yCTD variant treated with P-TEFb alone (red) and no kinase control reaction (black). Mass labels indicate $m/z$ at the various peak maxima. Arrows indicate the range of $m/z$ peaks for kinase treated sample. '+#P' notation indicates an approximate number of phosphates added based on mass shifts relative to no kinase control. (C) SDS-PAGE EMSA of dmCTD and dmCTDYtoF (as indicated) treated with Erk2 (top, right bands), TFIIH (middle, right bands), or P-TEFb (bottom, right bands) and paired no kinase control reactions (left bands). (D–E) Kinase activity assay of wild-type yCTD (shown in black) and yCTDYtoF (shown in red) variant by Erk2 (D) and TFIIH (E) fitted to the Michaelis-Menten kinetic equation. The Michaelis-Menten kinetic parameters $k_{cat}(s^{-1})$, $K_m(\mu M)$, and $k_{cat}/K_m(\mu M^{-1} s^{-1})$ are given below the graphs for each respective fit. Each measurement was conducted in triplicate with standard deviations shown as error bars.

DOI: https://doi.org/10.7554/eLife.48725.003

The following figure supplements are available for figure 1:

**Figure supplement 1.** Amino acid sequences of GST-CTD constructs.
DOI: https://doi.org/10.7554/eLife.48725.004

**Figure supplement 2.** Mass spectrometry analysis of yCTD treated by kinases.
DOI: https://doi.org/10.7554/eLife.48725.005

serine residues subject to phosphorylation. Several factors suggest the chemical properties of residues at the Tyr1 position are important for CTD modification. First, residues at this first position of the heptad are highly conserved across species and substitution to non-aromatic residues is rare, suggesting significance to function (*Chapman et al., 2008*). As evidence of this, even conservative mutation of the Tyr1 position to phenylalanine in both *Saccharomyces cerevisiae* and human cells is lethal, highlighting the significance of residue identity at this position (*Hsin et al., 2014*; *West and Corden, 1995*). Secondly, we have shown that mutating the Tyr1 position to alanine prevents phosphorylation at other CTD residues by CTD kinases (*Mayfield et al., 2017*), indicating the side chain at this position is important for kinase activity. Third, phosphoryl-Tyr1 (pTyr1) is detected at the initiation of transcription in human cells (*Descostes et al., 2014*). This positions pTyr1 well to influence and interact with subsequent modifications of the CTD and, potentially, direct subsequent enzyme specificities.

To determine the effect of the chemical characteristics of residues at the Tyr1 position on CTD modification, we searched for naturally occurring Tyr1 substitutions. *Drosophila melanogaster* CTD contains a majority of heptads that diverge from consensus sequence with only 2 of its approximately 45 heptads being of the consensus sequence. Despite this highly divergent character, the Tyr1 position of *D. melanogaster* CTD is rather conserved and contains mostly tyrosine residues. For the six heptads that do not contain tyrosine, half are modestly substituted with phenylalanine. We were curious to determine if, like alanine, phenylalanine replacement at the Tyr1 position would abolish CTD kinase activity. These initial experiments were designed on *D.* melanogaster CTD because its heptads have diverse sequences that allow for the observation of shifts in electrophoretic mobility shift assay (EMSA) banding patterns, which might not be easily seen for consensus sequence CTD substrates. We generated GST-CTD constructs containing a portion of *D. melanogaster* CTD (residues 1671–1733, containing nine heptad repeats) of either tyrosine containing wild-type (dmCTD) or with phenylalanine substitution at the Tyr1 position in all nine heptads (dmCTDYtoF) (*Figure 1—figure supplement 1D*). The CTD variants purified from these constructs were phosphorylated with one of three established CTD kinases: Erk2, a recently identified Ser5 CTD kinase that phosphorylates primed RNA polymerase II in developmental contexts (*Tee et al., 2014*); the kinase module of TFIIH that install Ser5 and Ser7 marks in vivo (*Feaver et al., 1994*); or P-TEFb which phosphorylates Ser2 in vivo (*Marshall et al., 1996*). Unlike alanine substitution, all three kinases are active against the phenylalanine-substituted CTD construct (*Figure 1C*). Surprisingly, the substitution of phenylalanine at the Tyr1 position alters the behavior of phosphorylated substrates in EMSA (*Figure 1C*). While the wild-type variant assumes only one or two apparent intermediate species in EMSA, the YtoF variant of dmCTD exhibits multiple intermediates, suggesting the generation of a greater diversity of phosphorylated species. Additional analysis of Erk2 phosphorylated dmCTDYtoF using electrospray ionization mass spectrometry (ESI-MS) of the intact phosphorylated construct confirms the existence of multiple species revealing complex spectra composed of multiple overlapping peaks relative to the dmCTD control (*Figure 1—figure supplement 2D*). To quantify the effect of phenylalanine replacement at the Tyr1 position on CTD kinase function, we measured the kinase activity of Erk2 and TFIIH using GST-yCTD or yCTDYtoF (in which all Tyr1 positions have been mutated to phenylalanine) substrates (*Figure 1—figure supplement 1A–B*). Steady-state kinetics demonstrate that the replacement of tyrosine by phenylalanine has a markedly different effect on these two kinases. Erk2 shows a 2.5-fold higher specificity constant against the YtoF variant, as indicated by $k_{cat}/K_m$, compared to the WT construct (*Figure 1D*). Erk2 has nearly identical $k_{cat}$ values ($0.44 \pm 0.02$ s$^{-1}$ vs. $0.45 \pm 0.02$ s$^{-1}$) for the two substrates, but a much lower $K_m$ for the YtoF substrate ($35.8 \pm 3.2$ μM for WT vs. $14.8 \pm 2.2$ μM for YtoF substrates). This difference in $K_m$ values suggests Erk2 has a binding preference for the phenylalanine substituted substrate. However, TFIIH activity is greatly compromised when Tyr1 is replaced by phenylalanine with a nearly 10-fold reduction in $k_{cat}/K_m$ (*Figure 1E*).

Overall, our data demonstrate the chemical properties of the residues located at the first position of the heptad repeat have a significant impact on the phosphorylation of the CTD by CTD kinases. Even slight modification of this residue (e.g., loss of the hydroxyl group) can have dramatic consequences for modification of the CTD.

## Tyr1 phosphorylation in human CTD

Although substitution of non-tyrosine residues at the Tyr1 position is relatively rare in nature and does not occur in human cells, Tyr1 phosphorylation is conserved from yeast to humans and plays a key role in transcriptional events (*Chapman et al., 2008*; *Yurko and Manley, 2018*). Since the molecular mechanism explaining its diverse biological functions remains elusive, the sensitivity of CTD kinases to the chemical properties of Tyr1 side-chain motivated us to investigate if Tyr1 phosphorylation impacts subsequent phosphorylation events by reconstructing sequential CTD phosphorylation in vitro. In humans, Tyr1 phosphorylation rises along with Ser5 phosphorylation at the beginning of transcription (*Heidemann et al., 2013*). However, experiments using synthetic CTD peptides with every Tyr1 residue phosphorylated have shown that Tyr1 phosphorylation inhibits subsequent phosphorylation by CTD kinases (*Czudnochowski et al., 2012*). We suspect that the heavily phosphorylated synthetic peptide doesn't mimic the physiological RNA polymerase II during transcription. Instead, we reconstructed the phosphorylation of the CTD using physiologically relevant Tyr1 kinases in vitro. Existing literature points to Abl-like non-receptor tyrosine kinases as mammalian Tyr1 CTD kinases, with c-Abl as a major candidate (*Baskaran et al., 1997*; *Burger et al., 2019*). Three lines of evidence support this notion: c-Abl phosphorylates CTD in vitro (*Baskaran et al., 1993*) and in cells since transient over-expression of c-Abl in primate COS cells results in increased Tyr1 phosphorylation (*Baskaran et al., 1997*), and c-Abl immunoprecipitates with RNA polymerase II (*Baskaran et al., 1999*). To elucidate the biophysical consequences of Tyr1 phosphorylation of the CTD, we reconstructed c-Abl phosphorylation of consensus sequence CTD in vitro using purified human c-Abl and the yCTD constructs. C-Abl readily phosphorylates yCTD in vitro as evidenced by EMSA and detection of Tyr1 phosphorylation using pTyr1 specific antibody 3D12 (*Figure 2A*). We directly interrogate the sites of phosphorylation using LC-UVPD-MS/MS to identify phosphorylation sites in single residue resolution (*Mayfield et al., 2017*). Using this method to analyze a peptide containing three heptad repeats treated by c-Abl (3CTD, *Figure 1—figure supplement 1E*), two single phospho-forms were detected, each containing a single phosphorylated tyrosine on either the first or second heptad of the variant (*Figure 2B* and *Figure 1—figure supplement 2E*). These mass shifts confirm c-Abl phosphorylates consensus CTD sequences on Tyr1 in vitro.

To test if modulating c-Abl activity can alter Tyr1 phosphorylation levels in human cells, we treated HEK293T cells with the c-Abl specific inhibitor imatinib (*Knight and McLellan, 2004*) and monitored endogenous Tyr1 phosphorylation levels using phospho-Tyr1 specific antibody 3D12 (*Figure 2C*). Imatinib has potent and specific inhibition against c-Abl and ABL2 (which shares 93% sequence identity in the kinase domain as c-Abl) (*Salah et al., 2011*). Tyr1 phosphorylation decreases in a dose-dependent manner from 10–50% at imatinib concentrations of 10–30 μM after 24 hr of treatment (*Figure 2C*). Overall, our result indicates controlling the kinase activity of c-Abl, or highly similar kinases, is sufficient to significantly modulate the level of Tyr1 phosphorylation of CTD in mammalian cells.

We next quantified the maximal number of phosphates added to yCTD constructs using MALDI-MS. High-resolution MALDI-MS spectra of samples treated by c-Abl revealed peaks accounting for yCTD containing 5 to 13 phosphates (with mass shifts ranging from 398.9 to 1049.4 Da) (*Figure 2D*), approximately half of yCTD's available tyrosine residues within 26 heptad repeats. Further incubation with more kinase/ATP does not appear to add more than 13 phosphate groups to substrate CTD in these assays. Additional GST-CTD constructs containing 3–5 consensus heptad repeats (*Figure 1—figure supplement 1E–G*) treated with c-Abl were analyzed using MALDI-TOF to evaluate if c-Abl truly only phosphorylates half of the available Tyr1 sites even in the presence of high kinase/ATP concentrations and after prolonged incubation times. Three phosphorylation peaks were detected in the 5CTD variant with mass differences of 79.8, 160.3, and 239.1 Da relative to the unphosphorylated control, accounting for the addition of 1–3 phosphates (*Figure 2E*). Phosphorylation of the 4CTD construct resulted in two peaks of phosphorylation with mass differences of 79.1 and 160.7 Da relative to unphosphorylated control, accounting for the addition of 1 or 2 phosphates (*Figure 2F*). Similarly, two phosphates are added to the 3CTD variant that displayed mass shifts of 78.2 or 158.7 Da (*Figure 2G*). These mass shifts suggest c-Abl does not phosphorylate consensus CTD in every heptad; instead, it favors phosphorylation of approximately half the available Tyr1 residues.

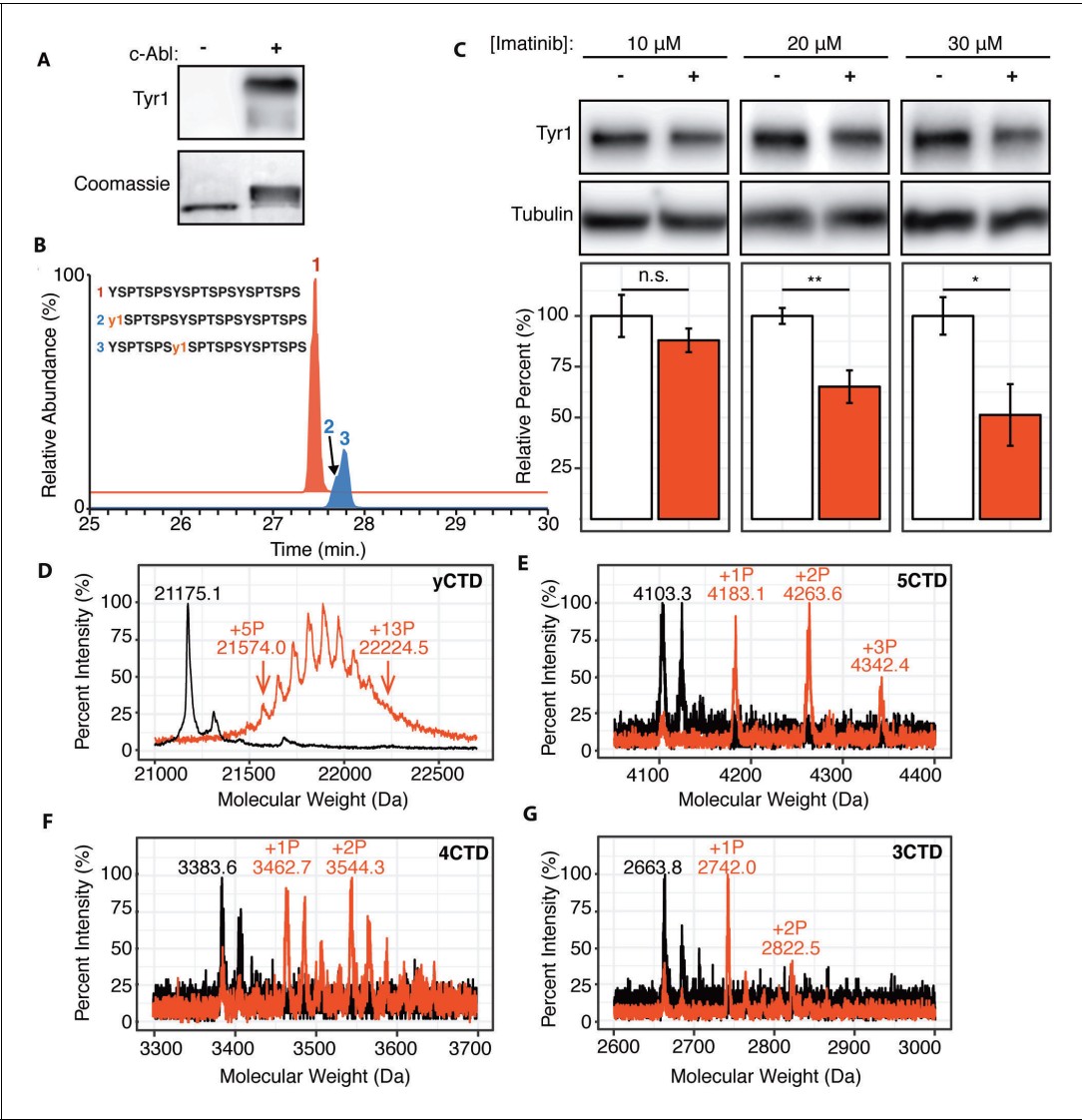

**Figure 2.** c-Abl kinase phosphorylates Tyr1 of RNA polymerase II CTD in cells and in vitro. (**A**) Representative image of western blot against phosphorylated Tyr1 (top) of yCTD (containing 26 heptad repeats) treated in vitro with c-Abl (right) and paired no kinase control (left). Coomassie-stained blot included indicating loading (bottom). Data representative of three experimental replicates. (**B**) LC-UVPD-MS/MS analysis of 3CTD treated with c-Abl showing extracted ion chromatograms for 3CTD (*m/z* 888.76, 3+ charge state, red trace) and mono-phosphorylated 3CTD (*m/z* 915.36, 3+ charge state, blue trace) peptides. (**C**) Representative images (top) and quantification (bottom) of a western blot of imatinib dosage series (10–30 μM, as indicated, red) and paired DMSO vehicle controls (left band, white) of 20 μg total protein from HEK293T cells. Phospho-specific Tyr1 antibody (clone 3D12) was used. Imatinib decreases pTyr1 epitope abundance to 88.0% (10μM imatinib, not significant), 65.2% (20μM imatinib), and 51.3% (30μM imatinib) relative to paired vehicle controls. Epitope signals normalized against tubulin loading control. Significance determined by Welch's t-test (*=p value<0.05, **=p value<0.01, n.s. = not significant (p-value>0.05)), n = 6, error bars indicate SEM. (**D–G**) Portions of MALDI mass spectra of 3C-protease digested yeast CTD (**D**), 5CTD (**E**), 4CTD (**F**), and 3CTD (**G**) construct treated with c-Abl (red) and no kinase control reaction (black). Mass labels indicate *m/z* at various peak maxima. Arrows indicate the range of *m/z* peak for kinase treated sample. '+#P' notation indicates an approximate number of phosphates added based on mass shifts. Satellite peaks, prevalent in (**F**), correlate well with sodium adducts (M+23 Da).
DOI: https://doi.org/10.7554/eLife.48725.006

## Sequential phosphorylation of CTD by c-Abl followed by P-TEFb

With our knowledge of the previously undescribed Tyr1 phosphorylation pattern installed by c-Abl, we were curious if such a pattern could affect the phosphorylation of CTD by P-TEFb. We first determined if the pre-treatment of CTD by c-Abl alters the number of phosphates added by P-TEFb using MALDI-TOF. Since c-Abl phosphorylates tyrosine and P-TEFb phosphorylates serine residues as

determined (*Figure 2A, B and 1A*, respectively), if the two phosphorylation events are independent, the number of phosphates placed by the two kinases should be additive. C-Abl phosphorylation of yCTD alone adds up to 13 phosphates (*Figure 2D*), and P-TEFb alone adds 24–29 phosphates (*Figure 1B*). Interestingly, tandem treatment of yCTD with c-Abl followed by P-TEFb resulted in the addition of a total of 16 to 26 phosphates as detected by MALDI-MS, with a mass shifts of 1287.3 to 2073.2 Da (*Figure 3A*). These data reveal c-Abl pre-treatment results in changes to P-TEFb's phosphorylation along the CTD, evidenced by a reduction in the number of phosphate groups added by P-TEFb.

To identify the position of phosphates added by P-TEFb when Tyr1 is phosphorylated, we quantified pSer2 and pSer5 by immunoblotting with antibodies recognizing Ser2 and Ser5 phosphorylations (*Figure 3B*). Compared to a non-phosphorylated CTD, the pre-treatment of yCTD with c-Abl results in a significant increase in Ser2 phosphorylation of nearly 300%, as detected by pSer2 specific CTD antibody 3E10, accompanied by a small and statistically non-significant decrease in pSer5 installed by P-TEFb (*Figure 3B and C*). The increase of pSer2 is unique for P-TEFb-mediated phosphorylation of CTD since a similar tandem treatment of yCTD by c-Abl followed by either TFIIH or Erk2 showed no changes in pSer2 levels (*Figure 3C* and *Figure 3—figure supplement 1A*).

We propose two possible explanations for the apparent increase of pSer2 levels upon c-Abl/P-TEFb treatment: First, c-Abl interacts with and/or modifies P-TEFb and alters its specificity from Ser5 to Ser2. Alternatively, c-Abl may phosphorylate substrate CTD and these phosphorylations prime P-TEFb specificity towards Ser2 residues of the CTD. To differentiate these two models, we inactivated c-Abl after its reaction with CTD but before the addition of P-TEFb. We used two independent methods to inactivate c-Abl prior to P-TEFb addition: introduction of the potent Abl inhibitor dasatinib to 10 µM or denaturation of c-Abl via heat-inactivation (*Figure 3—figure supplement 1B and C*). In the first method, the introduction of dasatinib inhibits c-Abl activity towards the CTD but shows no effect in P-TEFb's ability to phosphorylate the CTD substrate (*Figure 3—figure supplement 1C*). In the second method, the heat-inactivation effectively abolishes the kinase activity of c-Abl (*Figure 3—figure supplement 1B*). In both experiments, P-TEFb continues to install a greater amount of Ser2 phosphorylation relative to no c-Abl treatment controls (*Figure 3—figure supplement 1D*). Therefore, the increase in the apparent Ser2 phosphorylation is not due to P-TEFb's physical interaction with c-Abl but arises from c-Abl kinase activity against CTD substrates at Tyr1.

Since immunoblotting presents the issue of epitope masking in highly phosphorylated protein samples and lacks the ability to separate reaction products in high resolution, we sought a method to identify the phosphorylation sites on the CTD directly and precisely. To determine the phosphorylation pattern resulting from sequential kinase treatment, we used LC-UVPD-MS/MS to investigate the activity of P-TEFb in the context of Tyr1 phosphorylation. LC-UVPD-MS/MS provides single residue resolution and overcomes artifacts inherent to immunoblotting such as epitope masking. Unfortunately, full-length yeast CTD is resistant to proteolysis due to a lack of basic residues, hindering further analysis by tandem MS (*Schüller et al., 2016*; *Suh et al., 2016*). Novel proteases, such as chymotrypsin and proteinase K, that cleave at bulky hydrophobic residues like tyrosine have proven effective in the past for analyzing the native sequence of the CTD, but proteolysis becomes inhibited upon phosphorylation due to the modification on tyrosine (*Mayfield et al., 2017*). Short synthetic peptides circumvent the need for proteases but poorly mimic the physiological CTD and are unlikely to reveal bona fide CTD kinase specificities. To overcome these technical challenges, we generated a full-length yeast CTD with lysine replacing Ser7 in every other repeat (yCTD-Lys) (*Figure 1—figure supplement 1C*). This allowed for trypsin digestion into di-heptads, which represent the functional unit of the CTD (*Corden, 2013*; *Eick and Geyer, 2013*), and are amenable to MS/MS analysis.

To validate that the introduction of lysine residues does not bias kinase specificity, we first mapped the phosphorylation pattern of c-Abl or P-TEFb individually along yCTD-Lys using LC-UVPD-MS/MS. When treated with c-Abl, two single phosphorylation species are found at equivalent abundances with tyrosine at the same or neighboring heptad of Lys replacement (*Figure 3D* and *Figure 3—figure supplement 2A*, peak 1 and 2) and a small peak in which both Tyr1 residues are phosphorylated (*Figure 3D* and *Figure 3—figure supplement 2A*, peak 3). When treated with P-TEFb alone, we observed four single phosphorylated peptides: two almost equally abundant peaks containing di-heptads with a single Ser5 phosphorylation (*Figure 3E* and *Figure 3—figure supplement 2B*, peak 1 and 2) and two peaks about ~40 fold less in intensity with pSer2 or pSer7 (*Figure 3E* and *Figure 3—figure supplement 2B*, peaks 3 and 4). This result shows that P-TEFb

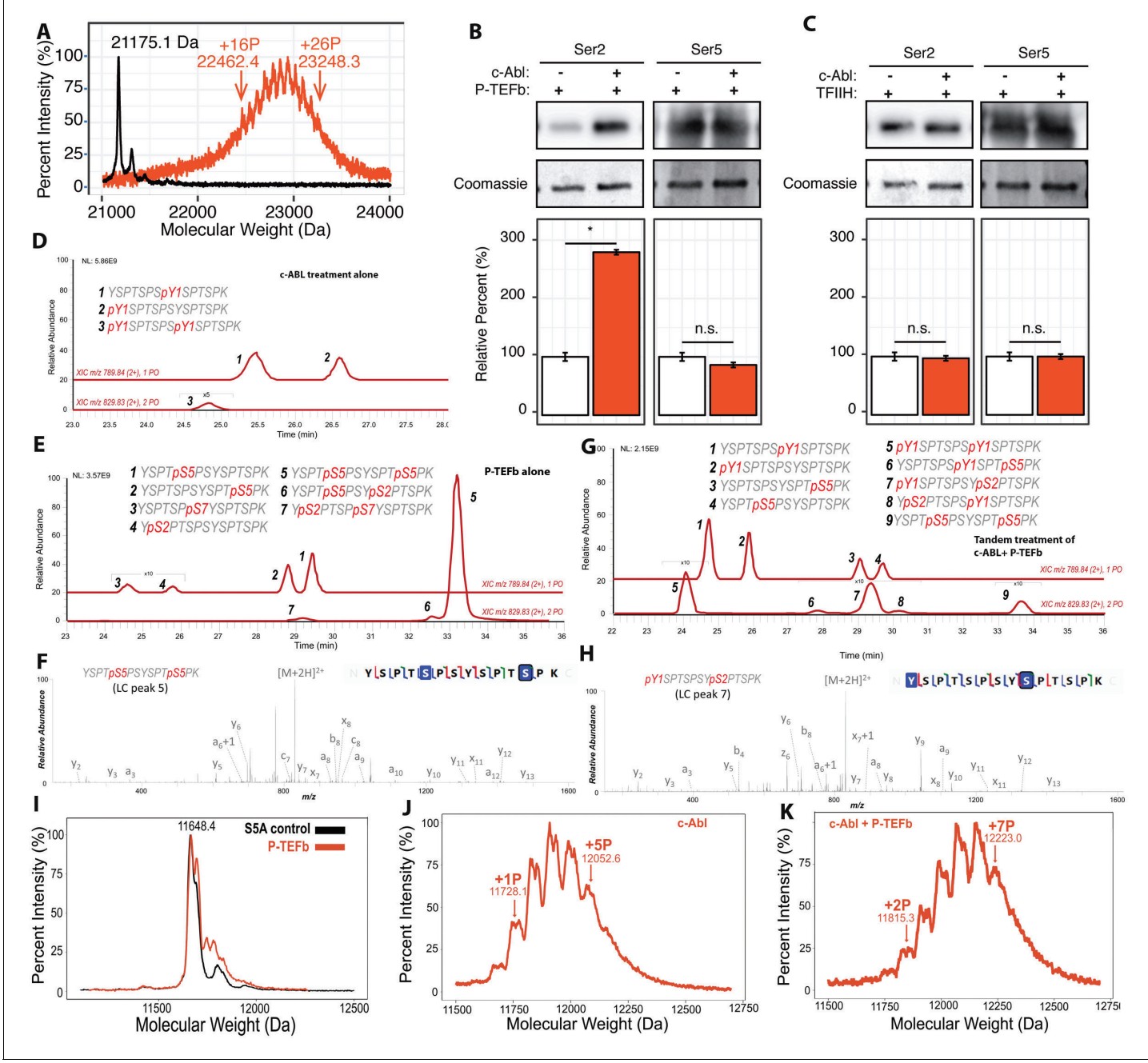

**Figure 3.** Effect of Tyr1 phosphorylation by c-Abl on the function of P-TEFb. (**A**) Portions of MALDI mass spectra of 3C-protease digested yCTD construct treated tandemly with c-Abl followed by P-TEFb (red) and no kinase control reaction (black). Mass labels indicate *m/z* peaks for kinase treated sample. '+#P' notation indicates an approximate number of phosphates added based on mass shifts. (**B**) Representative images (top) and quantification (bottom) of western blot analysis of yCTD treated with P-TEFb alone (left, white) and tandemly with c-Abl followed by P-TEFb (right, red). Tandem treatment of c-Abl followed by P-TEFb increases phosphorylated Ser2 epitope abundance to 279% of P-TEFb only treatment control. Ser5 phosphorylation levels are not significantly altered. Significance determined by Welch's t-test (*=p value<0.05, n.s. = not significant (p-value>0.05)), n = 3, error bars indicate SEM. (**C**) Representative images (top) and quantification (bottom) of western blot analysis of yCTD treated with TFIIH alone (left, white) and tandemly with c-Abl followed by TFIIH (right, red). Tandem treatment of c-Abl followed by TFIIH does not significantly alter the epitope abundance of phosphorylated Ser2 or Ser5. Significance determined by Welch's t-test (n.s. = not significant (p-value>0.05)), n = 3, error bars indicate SEM. (**D–H**) LC-UVPD-MS/MS analysis of yeast CTD with inserted Lys in every other heptad repeat (yCTD-Lys) treated with kinases or combination of kinases. Biological triplet samples were independently measured with exemplary spectra shown (n = 3). (**D**) yCTD-Lys treated with c-Abl showing extracted ion chromatograms for mono-phosphorylated (*m/z* 789.84, 2+ charge state) and doubly phosphorylated (*m/z* 829.83, 2+ charge state) peptides of sequence (YSPTSPSYSPTSPK). The LC traces are shown in red, and the phosphorylation sites determined by UVPD-MS/MS are highlighted

*Figure 3 continued on next page*

*Figure 3 continued*

in red font with 'p' to indicate phosphorylation. (**E**) LC-UVPD-MS/MS analysis of yCTD-Lys treated with P-TEFb showing extracted ion chromatograms for mono-phosphorylated (*m/z* 789.84, 2+ charge state) and doubly phosphorylated (*m/z* 829.83, 2+ charge state) peptides. (**F**) Representative UVPD spectra that demonstrate the diagnostic fragmentation pattern of the peptides shown in the inset from (**E**). The one shown is peak 5, which is the predominant product from yCTD-Lys treated by P-TEFb alone. (**G**) LC-UVPD-MS/MS analysis of yCTD-Lys with c-Abl followed by P-TEFb showing extracted ion chromatograms for mono-phosphorylated (*m/z* 789.84, 2+ charge state) and doubly phosphorylated (*m/z* 829.83, 2+ charge state) peptides. (**H**) Representative UVPD spectra that demonstrate the diagnostic fragmentation pattern of the peptides shown in the inset of (**G**). The one shown is peak 7, which is the predominant product from yCTD-Lys treated by c-Abl followed by P-TEFb. (**I–K**) Portions of MALDI-MS spectra of 3C-digested S5A. Panels included no kinase control (**I**), P-TEFb only treated (**J**), c-Abl only treated (**I**), and tandemly treated with c-Abl followed by P-TEFb (**K**). Mass labels indicate *m/z* at the various peak maxima. Arrows indicate the maximum and minimum *m/z* peak for kinase treated sample. '+#P' notation indicates an approximate number of phosphates added based on mass shifts.

DOI: https://doi.org/10.7554/eLife.48725.007

The following figure supplements are available for figure 3:

**Figure supplement 1.** Phosphorylation of different CTD constructs by CTD kinases.

DOI: https://doi.org/10.7554/eLife.48725.008

**Figure supplement 2.** LC-UVPD-mass spectra yCTD-Lys treated with CTD kinases.

DOI: https://doi.org/10.7554/eLife.48725.009

**Figure supplement 3.** LC-UVPD-mass spectra yCTD-Lys treated with CTD kinases.

DOI: https://doi.org/10.7554/eLife.48725.010

**Figure supplement 4.** Analysis of yCTD treated in tandem with TFIIH and P-TEFb.

DOI: https://doi.org/10.7554/eLife.48725.011

strongly favors pSer5 in unmodified CTD substrates, consistent with our previously analysis (*Figure 1A*). Double phosphorylated species are also detected for di-heptads with both Ser5 residues phosphorylated as the predominant product (*Figure 3E and F* and *Figure 3—figure supplement 2B*, peak 5). Several very small peaks (less than 100-fold lower in intensity) are identified as peptides containing both Ser5 and pSer2 (*Figure 3E* and *Figure 3—figure supplement 2B*, peak 6). These results indicate that the existence of Lys residue does not seem to bias kinase activity and is consistent with our previous results that P-TEFb strongly prefers to phosphorylate Ser5.

When treated in tandem with c-Abl followed by P-TEFb and digested with trypsin, di-heptads (YSPTSPSYSPTSPK) in a variety of phosphorylation states are generated. These species were separated in liquid chromatography (LC) and revealed nine di-heptad species of varying abundances. LC purification separates the different phosphorylation states of the di-peptide (*Figure 3G*). Some of the di-heptads contain only single phosphorylations due to incomplete reactions in vitro. To understand the effect of c-Abl CTD phosphorylation on P-TEFb, we focused on phosphorylated species with more than one phosphate added, especially those containing both tyrosine and serine phosphorylation (*Figure 3G*). Tandem phosphorylation generated species unique to those observed in c-Abl or P-TEFb individual treatment (elution at 27–31 min in LC, *Figure 3G*, peak 6, 7, 8). The most abundant of these unique species (*Figure 3G* peak 7) contains both Tyr1 and Ser2 phosphorylation (*Figure 3H* and *Figure 3—figure supplement 2C*). Similarly, a close-by but less abundant peak also contains Tyr1 and Ser2 double phosphorylation although in a different location (*Figure 3G* peak eight and *Figure 3—figure supplement 2C*). Only a small peak contains both Tyr1 and Ser5 phosphorylation (*Figure 3G* peak six and *Figure 3—figure supplement 2C*). Although we cannot exclude the possibility of the existence of other phosphorylated species containing a mixture of tyrosine and serine phosphorylation, their quantity is likely very low and not detected in LC-UVPD-MS/MS analysis. P-TEFb's serine residue preference is dramatically different between reactions on unmodified CTD substrate, where pSer5 predominates (*Figure 3E* peak 5), and those pre-treated with c-Abl where pSer2 is the primary product species (*Figure 3G* peak 7). Our results show that in di-heptads with Tyr1 phosphorylated, Ser2 becomes the primary target of P-TEFb phosphorylation.

The high performance of LC chromatography also allowed us to confirm our phosphomapping within di-heptads of the yeast CTD that diverge from the consensus sequence (*Figure 1—figure supplement 1C*). Three di-heptads of the divergent sequence were generated following trypsin digestion of the yCTD-Lys construct (*Figure 3—figure supplement 3*, sequences of YSPT-SP<u>A</u>YSPTSPK, YSPTSP<u>N</u>YSPTSPK, and YSPTSP<u>G</u>YSP<u>G</u>SPK). Although these di-heptads exist in a much smaller amount than the dominant product YSPTSP<u>S</u>YSPTSPK, they can be resolved and

purified in high-performance liquid chromatography and analyzed for phosphorylation position (*Figure 3—figure supplement 3*). In these three phosphorylated di-heptads, the sole detected product of tandem treatment is a di-heptad with Tyr1 and Ser2 phosphorylated (*Figure 3—figure supplement 3*, right panels). In contrast, all peptides phosphorylated by P-TEFb alone gave a predominant di-heptad species containing only pSer5 (*Figure 3—figure supplement 3*, left panels). No Tyr1 and Ser5 double phosphorylation species were captured, possibly due to low abundance. The phosphoryl mapping of the various di-heptads generated provides independent evidence that Tyr1 phosphorylation promotes Ser2 phosphorylation by P-TEFb even in the context of divergent heptads.

To further corroborate the mass spectrometry results that the specificity of P-TEFb is altered from Ser5 to Ser2 upon Tyr1 phosphorylation, we generated a new yeast CTD variant with 13 repeats (half of the full-length yeast CTD) with every single Ser5 mutated to alanine (S5A construct, *Figure 1—figure supplement 1H*). Previously, it was shown that replacing Ser5 in CTD prevents its phosphorylation by P-TEFb (*Czudnochowski et al., 2012*). Treatment of the S5A constructs with P-TEFb alone results in the addition of little to no phosphate shown by MALDI-MS (*Figure 3I*). However, when the S5A construct is treated with c-Abl, it accepts up to five phosphate groups (*Figure 3J*). Subsequent treatment with P-TEFb results in an obvious shift in the MALDI-MS spectra with up to seven phosphates added to the final product (*Figure 3K*). The results corroborate the conclusion drawn from the MS/MS results, indicating that upon Tyr1 phosphorylation S5A becomes a viable substrate for P-TEFb and adds at least two additional phosphates likely to Ser2 residues.

The observation of pSer2 as the major product in the context of pre-existing pTyr1 is interesting because P-TEFb has consistently shown a strong preference for Ser5 in vitro. Using a combination of immunoblotting, LC-UVPD-MS/MS, mutagenesis, and MALDI-MS we found Tyr1 phosphorylation primes the CTD for subsequent modification on Ser2 by P-TEFb via alteration of its specificity from Ser5.

## Effect of Ser5 and Ser7 phosphorylation on P-TEFb activity

The observation that Tyr1 phosphorylation by c-Abl alters the specificity of P-TEFb from Ser5 to Ser2 prompted us to ask if other kinases recruited to the CTD at the beginning of transcription can also alter P-TEFb specificity in vitro. TFIIH is a kinase that acts during transcription initiation and promotes P-TEFb function in vivo (*Ebmeier et al., 2017*). To evaluate if a modification or combination of modifications installed by TFIIH can promote the Ser2 specificity of P-TEFb as we see with Tyr1 phosphorylation, we reconstructed CTD phosphorylation in vitro by treating yCTD substrates sequentially with TFIIH followed by P-TEFb and analyzed the resultant phosphorylation pattern using LC-UVPD-MS/MS and immunoblotting (*Figure 3—figure supplement 4*). When followed by P-TEFb, three phosphorylated species are generated, as revealed by LC-UVPD-MS/MS: two major species containing Ser5 phosphorylation and a minor species containing Ser2 phosphorylation (*Figure 3—figure supplement 4A–B*). These peptides are reminiscent of those generated by P-TEFb alone where Ser5 phosphorylation dominates (*Figure 1A*). These data indicate that TFIIH-mediated phosphorylations do not alter P-TEFb specificity in vitro.

## Tyr1 phosphorylation primes Ser2 phosphorylation in human cells

Our kinase assays have shown that Tyr1 phosphorylation by c-Abl alters the specificity of P-TEFb allowing for Ser2 phosphorylation of the CTD in vitro. To evaluate the importance of pTyr1 to Ser2 phosphorylation in human cells, we sought to selectively reduce pTyr1 levels and monitor pSer2 via western blot (*Figure 4A and B*). Available literature suggests that c-Abl is important to Tyr1 phosphorylation in RNA polymerase II but not the sole kinase responsible (*Baskaran et al., 1999*). Other Abl-like kinases may likely compensate for the function of c-Abl by phosphorylating Tyr1 in human cells (*Baskaran et al., 1997*). Therefore, we initially utilized the potent inhibitor dasatinib, which inhibits c-Abl as well as other tyrosine kinases similar to c-Abl, to treat HEK293T cells (*Winter et al., 2012*). Tyr1 phosphorylation has also been implicated in stabilizing RNA polymerase II in the cytosol (*Hsin et al., 2014*), so marked reduction of Tyr1 phosphorylation may lead to a decrease in the global level of RNA polymerase II resulting in an apparent decrease in CTD phosphorylation levels. To address this potential artifact, we optimized inhibitor concentration to a level at which global RNA polymerase II levels are not significantly altered as determined by immunoblotting against RNA polymerase II subunits POLR2A and POLR2C (*Figure 4* and *Figure 4—figure supplement 1A*). At

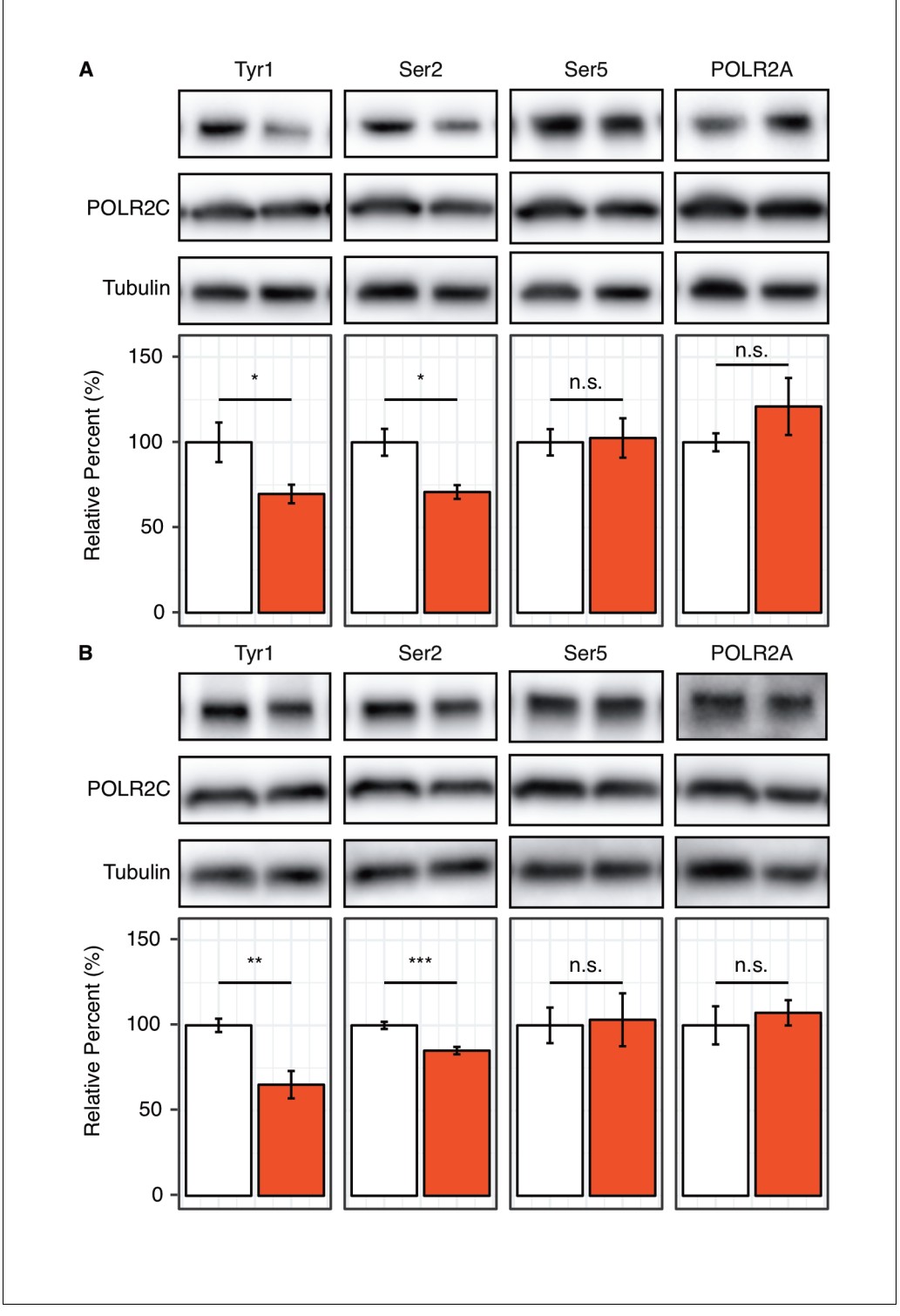

**Figure 4.** Reduction of Tyr1 levels specifically reduces Ser2 levels in cells. (**A**) Representative images (top) and quantification (bottom) of western blot against 20 µg total protein from HEK293T cells treated with paired DMSO vehicle control (left, white) or 10 µM dasatinib (right, red). Immuno-blotting against epitopes left to right: phosphorylated Tyr1 reduced by treatment to 69.7% control (n = 6), phosphorylated Ser2 reduced by treatment to 70.8% control (n = 4), phosphorylated Ser5 unaltered (n = 6), POLR2A unaltered (n = 6) (POLR2C quantification provided in *Figure 4—figure supplement 1A*). (**B**) Representative images (top) and quantification (bottom) of western blot against 20 µg total protein from HEK293T cells treated with paired DMSO vehicle control (left, white)
*Figure 4 continued on next page*

*Figure 4 continued*
or 20 µM imatinib (right, red). Immuno-blotting against epitopes, left to right: Phosphorylated Tyr1 reduced by treatment to 65.2% control (n = 6), phosphorylated Ser2 reduced by treatment to 85.2% control (n = 6), phosphorylated Ser5 unaltered (n = 6), POLR2A unaltered (n = 6) (POLR2C quantification supplied in *Figure 4— figure supplement 1A*). Epitope signals normalized against tubulin loading control. Significance determined by Welch's t-test (*=p value<0.05, **=p value<0.01, ***=p value<0.001, n.s. = not significant (p-value>0.05)), error bars indicate SEM.
DOI: https://doi.org/10.7554/eLife.48725.012
The following figure supplement is available for figure 4:

**Figure supplement 1.** Cell-based analysis of phosphorylation of RNA polymerase II.
DOI: https://doi.org/10.7554/eLife.48725.013

10 µM dasatinib, pTyr1 levels are reduced by 30% in HEK293T cells, and this is accompanied by a 29% decrease in Ser2 phosphorylation (*Figure 4A*). Importantly, pSer5 levels were not significantly altered (*Figure 4A* and *Figure 4—figure supplement 1A*). To more specifically target Abl-mediated Tyr1 phosphorylation, we utilized the highly specific inhibitor imatinib that has a much smaller inhibitory repertoire with strong inhibition to c-Abl and Abl2 (*Winter et al., 2012*). Treatment of HEK293T cells with 20 µM imatinib results in a reduction in pTyr1 of 35%. This is accompanied by a statistically significant decrease in pSer2 levels of 15% (*Figure 4B*). One potential concern is antibody masking by flanking phosphorylation because Tyr1 phosphorylation can block recognition of pSer2 by the 3E10 antibody (*Chapman et al., 2007*). Thus, loss of pTyr1 by inhibition with small molecules should produce an increase in the pSer2 signal if pSer2 levels remain constant. The fact that we observe a significant reduction in the pSer2 signal suggests it is indeed decreasing, but the 15–29% reduction quantified is likely an underestimate of the decrease given existing knowledge about these antibodies. Compounded with the mass spectrometry results, our data support that pTyr1 promotes Ser2 phosphorylation. In both the dasatinib and imatinib treatments pSer5, POLR2A, and POLR2C levels remain unaffected (*Figure 4* and *Figure 4—figure supplement 1A*). Data from this inhibitor-based approach are in line with our in vitro observation that Ser2 phosphorylation is specifically coupled to Tyr1 phosphorylation and extends these conclusions to cellular contexts.

## Decreased Tyr1 phosphorylation results in promoter proximal accumulation of RNA polymerase II

To understand the biological implication of coupled Tyr1 and Ser2 phosphorylations at the level of individual genes, we conducted ChIP-seq analysis for the distribution of RNA Polymerase II upon the inhibition of Tyr1 phosphorylation. To carry out this experiment, we inhibited c-Abl with the potent small-molecule inhibitor dasatinib in HEK293T cells under conditions where pTyr1 is significantly reduced, but overall Pol II amount is unaffected (*Figure 4A* and *Figure 4—figure supplement 1A*). The sample was prepared for ChIP-seq studies using RNA polymerase II antibody (8WG16) for immunoprecipitation to analyze the distribution of RNA polymerase II in a genome-wide fashion. In comparison of the dasatinib treated cells with the vehicle controls, the distribution of RNA polymerase II along the gene body is altered in multiple genes (*Figure 5A*). Signal was normalized to 10M reads for both samples, and a significant increase of peak height for RNA polymerase II was found in the promoter region of many genes, as demonstrated in representative genes Myc and FANCL (*Figure 5A*). To quantify the change of distribution of the polymerase, we calculated the pausing index (*Zeitlinger et al., 2007*) which is the ratio of Pol II read density in the region −50 to +300 bp of Transcription starting site (TSS) to the rest of gene body 3000 bp downstream of Transcription end site (TES). The genes are clustered into four groups based on the pausing index: the G0 cluster has a pausing score close to 0; the remaining genes were ranked based on their pausing scores from high to low with the G1 cluster containing genes with pausing scores less than the lower quartile, G2 with genes contained between the lower and upper quartile, and G3 with pausing scores above the upper quartile. Genes in G0 and G1 have little occupancy of the polymerase and might not be active (*Figure 5B*). A meta-analysis, as visualized in box plot for the pausing index of the genes in G0 and G1, shows no statistical difference between control and treatment samples (*Figure 4—figure supplement 1B and D*). But a statistically significant increase can be observed close to two-fold in G2 and G3 genes (*Figure 5C* and *Figure 4—figure supplement 1C–D*). The same trend is observed

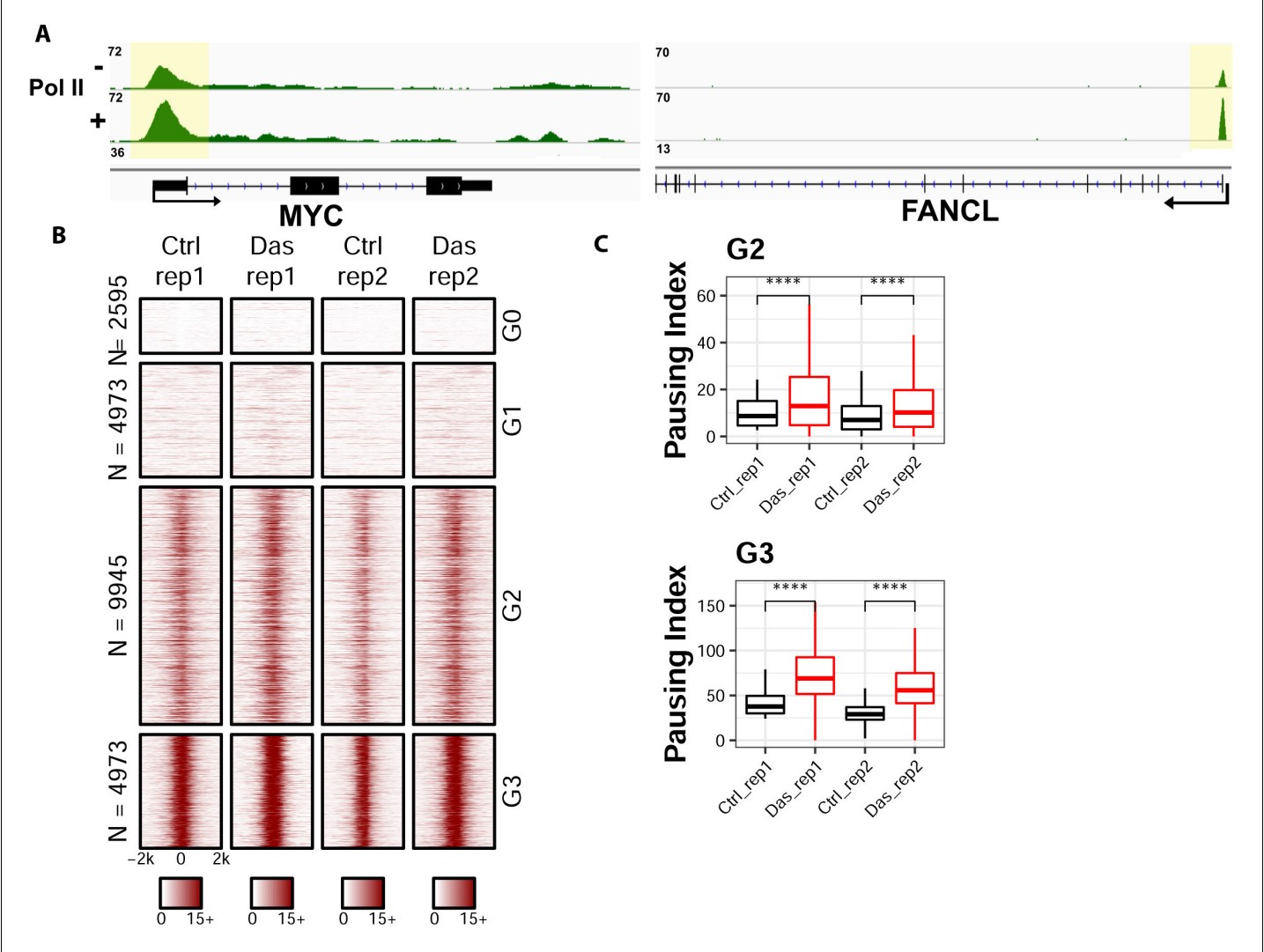

**Figure 5.** ChIP-seq analyses on the distribution of RNA polymerase II upon the inhibition of Tyr1 phosphorylation. (A) ChIP-seq example illustrating the association of RNA polymerase II along with the active transcribing genes. Antibody 8WG16 was used to detect RNA polymerase II regardless of its phosphorylation state. The promoter regions of the genes are shaded in yellow for highlighting. (B) Heatmaps of ChIP-seq signal intensity of RNA polymerase II (8WG16) [±2 kb windows around the center of transcription start site (TSS)] for genes in each group. (C) Boxplots on the pausing index changes on the genes from G2 (9945 genes) and G3 (4973 genes) clusters upon Tyr1 phosphorylation. '****' indicates p-value≤0.0001.

DOI: https://doi.org/10.7554/eLife.48725.014

across biological duplicates. Overall, these results suggest that RNA polymerase II is stalled in the promoter region upon the inhibition of Tyr1 phosphorylation.

## Discussion

Our discovery that Tyr1 phosphorylation of the CTD alters the preference of P-TEFb from Ser5 to Ser2 resolves the controversy surrounding P-TEFb's specificity (*Bartkowiak et al., 2010*; *Czudnochowski et al., 2012*). P-TEFb was initially identified as a CTD kinase that controls the elongation potential of RNA polymerase II, is required for the majority of RNA polymerase II transcription, and is specific for Ser2 in vivo (*Chao and Price, 2001*; *Marshall et al., 1996*; *Ni et al., 2004*). However, these early conclusions are at odds with in vitro data demonstrating P-TEFb is incapable of phosphorylating Ser2 of CTD peptides in vitro (*Czudnochowski et al., 2012*). Two other kinases, CDK12 and CDK13, display Ser2 kinase activity in cells but do not seem to play a major role in Ser2

phosphorylation in early transcriptional events (*Bartkowiak et al., 2010*; *Chen et al., 2007*). Investigations on the effect of CTD phosphorylations on P-TEFb specificity have revealed that Ser7 (*Czudnochowski et al., 2012*) and Ser5 (this manuscript) do not alter its preference for Ser5. Using direct methods, like mass spectrometry confirmed by immunoblotting and EMSA, we identified that Tyr1 phosphorylation could alter the specificity of P-TEFb from Ser5 to Ser2 in vitro. It should be stressed that continuous heptad repeats with phosphorylated pTyr1 inhibit subsequent CTD modification by P-TEFb (*Czudnochowski et al., 2012*). However, when treated biochemically with c-Abl, the pTyr1 pattern not only allows for P-TEFb phosphorylation but also shift the substrate preference from Ser5 to Ser2. Furthermore, inhibition of Tyr1 phosphorylation leads to the reduction of Ser2 phosphorylation in human cells and the accumulation of Pol II in the promoter-proximal pausing stage of transcription. Therefore, we show that Tyr1 phosphorylation potentiates Ser2 phosphorylation of the CTD by altering P-TEFb specificity.

The PTM state of the CTD has been correlated to the progression of transcription (*Jeronimo et al., 2016*). Traditionally, such analyses are interpreted through a paradigm considering heptads phosphorylated on a single isolated residue with Ser5 phosphorylation dominating the initiation stage of transcription and Ser2 phosphorylation dominating elongation and termination (*Corden, 2013*). However, this simplification of CTD modification cannot explain the well-coordinated recruitment of the myriad CTD binding factors currently implicated in eukaryotic transcription (*Ebmeier et al., 2017*; *Eick and Geyer, 2013*; *Harlen and Churchman, 2017*). Data presented here point to a sophisticated model in which the phosphorylation of Tyr1 at the beginning of transcription sets the stage for future coding events. This interplay between c-Abl and P-TEFb results in a chemically distinct phospho-CTD landscape compared to CTD phosphorylated by a single kinase. The combination of these modification modes likely contributes to a heterogeneous collection of modified heptads, which recruits the diverse array of CTD binding partners in a coordinated manner. These results are in good agreement with the 'CTD code' hypothesis proposed decades ago where different combinations of post-translational events result in different transcriptional outcomes.

Tyr1 phosphorylation has been implicated in stabilizing RNA polymerase II in cells (*Hsin et al., 2014*), transcription termination (*Mayer et al., 2012*) and anti-sense transcription (*Descostes et al., 2014*) but a coherent molecular basis for these disparate functions remains elusive. Our analysis provides a molecular mechanism demonstrating how Tyr1 phosphorylation can affect subsequent phosphorylation events carried out by other CTD kinases. The ability of Tyr1 phosphorylation to redirect signaling and influence subsequent modifications along the CTD, as revealed for P-TEFb, suggests these various roles for pTyr1 may arise indirectly. It can function through its impact on downstream CTD modifiers, highlighting integrated, indirect, and context-specific mechanisms for pTyr1 during co-transcriptional signaling. The final accumulation of individual species is dependent on the dynamic interplay of CTD kinases and phosphatases throughout the transcription cycle. Tyr1 phosphorylation is relatively transient, appearing at the transition from initiation to elongation and decreasing rapidly through the action of phosphatase(s) (*Eick and Geyer, 2013*). Despite this transient nature, pTyr1 is positioned in a vital window to alter P-TEFb specificity and regulate its phosphorylation pattern along RNA polymerase II. The adjustability of P-TEFb specificity by nearby Tyr1 phosphorylation reveals a novel mechanism for the regulation of P-TEFb kinase activity. With many binding partners in cells for P-TEFb, there might be additional regulators promoting the pSer2 activity of P-TEFb independent of or cooperatively with Tyr1 phosphorylation.

The data presented reconcile P-TEFb's in vitro and in vivo specificity and inspires new queries fundamental to CTD biology. P-TEFb is ubiquitously important for transcription across eukaryotic cells and often co-opted in disease states like HIV infection and cancer (*Franco et al., 2018*). The integrated CTD code revealed here represents a unique mechanism to manipulate P-TEFb and potentially other CTD modifiers. Future inquiries using similar multi-disciplinary approaches will hopefully reveal CTD modification patterns in greater detail at different stages of the transcription process in single amino acid resolution. Information such as this will define the temporal and spatial signaling allowing for the recruitment of transcriptional regulators during active transcription. Overall, our findings support a model in which cross-talk between CTD modification enzymes increases the diversity and coding potential of CTD heptads. This expands the lexicon of phosphorylation marks and can provide more specific recruitment of transcription regulators allowing for the precise control of eukaryotic transcription.

# Materials and methods

**Key resources table**

| Reagent type (species) or resource | Designation | Source or reference | Identifiers | Additional information |
|---|---|---|---|---|
| Cell line | HEK293T | ATCC | | |
| Antibody | Anti-Tyr1 (Clone 3D12) (rat monoclonal) | Millipore | Cat# MABE350 | (1:1000) |
| Antibody | Anti-beta Tubulin (rabbit polyclonal) | Abcam | Cat# ab6046, RRID:AB_2210370 | (1:10000) |
| Antibody | Anti-Ser2 (Clone 3E10) (rat monoclonal) | Millipore | Cat# 04–1571, RRID:AB_11212363 | (1:10000) for in vittro phosphorylated samples, (1:5000) for cell lysates and dot blot |
| Antibody | Anti-Ser5 (Clone 3E8) (rat monoclonal) | Millipore | Cat# 04–1572, RRID:AB_10615822 | (1:10000) for in vitro phosphorylated samples, (1:5000) for cell lysates |
| Antibody | Anti-POLR2A (Clone 4F8) (rat monoclonal) | Millipore | Cat# 04–1569, RRID:AB_11213378 | (1:5000) |
| Antibody | Anti-POLR2C (rabbit monoclonal) | Abcam | Cat# ab182150 | (1:5000) |
| Antibody | Goat Anti-Rat IgG Antibody HRP conjugate | Millipore | Cat# AP136P, RRID:AB_91300 | (1:20000) |
| Antibody | Goat Anti-Rabbit IgG H and L | Abcam | Cat# ab6721, RRID:AB_955447 | (1:20000) |
| Antibody | Anti-RNA polymerase II CTD repeat YSPTSPS antibody [8WG16] (mouse monoclonal) | Abcam | Cat# ab817, RRID:AB_306327 | 5 µl/Chip |
| Antibody | RNA Pol II Ser2-P antibody [3E10] (rat monoclonal) | Chromotek | RRID: AB_2631403 | 500 µl/Chip |
| Recombinant DNA reagent | 3CTD, 4CTD, 5CTD and Y to F gene fragments | IDT | | Cloned into pET28a derived vector with His GST tag |
| Recombinant DNA reagent | pET-His6-ERK2 MEK1_R4F _coexpression vector | Gift from Melanie Cobb | Addgene plasmid Cat#39212 | |
| Recombinant DNA reagent | S5A 13 repeat CTD | Biomatik | | Cloned into pET28a derived vector with His GST tag |
| Recombinant DNA reagent | S7K spaced CTD | Biomatik | | Cloned into pET28a derived vector with His GST tag |
| Recombinant DNA reagent | human ABL1 kinase domain (residues 229–511) | Kind gift from Kuriyan Lab | | |
| Peptide, recombinant protein | TFIIH(Cdk7/Cyclin H/MAT1 (CAK complex)) | Millipore | Cat# 14–476 | Used at a concentration of 0.025 µg/µl |
| Peptide, recombinant protein | P-TEFb (Cdk9/ Cyclin T1) | Millipore | Cat# 14–685 | Used at a concentration of 0.0075 µg/µl |

*Continued on next page*

*Continued*

| Reagent type (species) or resource | Designation | Source or reference | Identifiers | Additional information |
|---|---|---|---|---|
| Peptide, recombinant protein | c-Abl kinase | ProQinase | Cat# 0992-0000-1 | Used at a concentration of 0.0035 µg/µl |
| Commercial assay or kit | Pierce BCA Protein Assay Kit | Thermo Fischer Scientific | 23252 | |
| Commercial assay or kit | NEBNext Ultra II DNA Library Prep Kit for Illumina | NEB | E7645S | |
| Commercial assay or kit | NEBNext Multiplex Oligos for Illumina (Index Primers Set 1) | NEB | E7335S | |
| Chemical compound, drug | Imatinib | Selleck Chemicals | S1026 | 20 µM concentration |
| Chemical compound, drug | Dasatinib | Sigma Aldrich | CDS023389 | 10 µM concentration |
| Chemical compound, drug | 10 nCi/µl radiolabeled ATP | Perkin Elmer | NEG002A 100UC | |
| Chemical compound, drug | 0.45 µm nitrocellulose filters | Sigma Aldrich | WHA10401114 | |
| Chemical compound, drug | Econo-Safe Economical Biodegradable Counting Cocktail | Research Products International | SKU: 111175 | |
| Chemical compound, drug | HALT protease and phosphatase inhibitor cocktail | Thermo Fischer Scientific | Cat# 78440 | |
| Chemical compound, drug | Ribonuclease A | VWR lifesciences | CAS# 9001-99-4 | |
| Chemical compound, drug | Proteinase K | Ambion | Cat# 2542 | |
| Chemical compound, drug | Glycogen | Thermofischer scientific | Cat# R0561 | |
| Chemical compound, drug | 16% Formaldehyde solution (w/v), Methanol-free | Thermo scientific | Ref# 28908 | |
| Chemical compound, drug | SuperSignal West Pico Chemiluminescent Substrate | Pierce | 34079 | |
| Software, algorithm | ggplot2, R smoothing package | R-Studio | https://www.rstudio.com/ | |
| Software, algorithm | DataExplorer (AB) | Matrix Science | http://www.matrixscience.com/help/instruments_data_explorer.html | |
| Software, algorithm | Image J | NIH | https://imagej.nih.gov/ij/download.html | |
| Software, algorithm | XCalibur Qual Browser | Thermo Fischer Scientific | XCALI-97617 | |

*Continued*

| Reagent type (species) or resource | Designation | Source or reference | Identifiers | Additional information |
|---|---|---|---|---|
| Software, algorithm | ProSight Lite | Proteomics Center of Excellence Northwestern University | http://prosightlite.northwestern.edu/ | |
| Other | Ni-NTA | Qiagen | 30210 | |
| Other | Dynabeads Protein G | ThermoFischer Scientific | Cat# 10004D | |
| Other | AMPure XP beads | Beckman Coulter | Ref# A63881 | |
| Other | Vivaspin | Sartorius | VS2002 | |
| Other | Picofrit 75 µm id analytical columns | New Objective | | |
| Other | Picofrit 75 µm id analytical columns | New Objective | | |
| Other | Waters Xbridge BEH C18 | Milford | | |
| Other | Orbitrap Fusion Lumos Tribrid mass spectrometer | Thermo Fischer Scientific | | |
| Other | Velos Pro dual linear ion trap mass spectrometer | Thermo Fischer | | |
| Other | G:BOX imaging systems | Syngene | | |

## Protein expression and purification

CTD coding sequences (*Figure 1—figure supplement 1*) were subcloned into pET28a (Novagene) derivative vectors encoding an N-terminal His-tag a GST-tag and a 3C-protease site to generate GST-CTD constructs as described previously (*Mayfield et al., 2017*). 3CTD-5CTD and YtoF variants of CTD coding portions (*Figure 1—figure supplement 1*) were amplified from synthetic DNA templates generated by IDT. The S5A variant DNA and the S7K spaced DNA constructs were purchased from Biomatik as synthetic genes, amplified and subsequently cloned into the pET28a derivative vector described above. *Homo sapiens* Erk2 was expressed from pET-His6-ERK2-MEK1_R4F_coexpression vector as a gift from Melanie Cobb (Addgene plasmid #39212) (*Khokhlatchev et al., 1997*). *E. coli* BL21 (DE3) cells grown 37°C in Luria-Bertani (LB) media were used to overexpress recombinant GST-CTDs variants and Erk2.

## GST-CTD variants were prepared by established protocols.

Briefly, proteins were overexpressed in *E. coli* BL21 (DE3) cells by growing at 37°C in LB media containing 50 µg/mL kanamycin to an $OD_{600}$ of 0.4–0.6. Expression was induced by the addition of isopropyl-β-D-thiogalactopyranoside (IPTG) to a final concentration of 0.5 mM. After induction, the cultures were grown at 16°C for an additional 16 hr. The cells were pelleted and lysed via sonication in lysis buffer [50 mM Tris-HCl pH 8.0, 500 mM NaCl, 15 mM Imidazole, 10% Glycerol, 0.1% Triton X- 100, 10 mM β-mercaptoethanol (BME)]. The lysate was cleared by centrifugation, and the supernatant was initially purified using Ni-NTA (Qiagen) beads and eluted with elution buffer (50 mM Tris-HCl pH 8.0, 500 mM NaCl, 200 mM Imidazole, and 10 mM BME). The protein was dialyzed against gel filtration buffer (20 mM Tris-HCl pH 8.0, 50 mM NaCl, 10 mM BME for GST-yCTD and 20 mM Tris-HCl pH 7.5, 200 mM NaCl, 10 mM BME). Finally, proteins were concentrated and ran on a Superdex 200 gel filtration column (GE). Erk2 was purified using a previously published protocol (*Khokhlatchev et al., 1997*). Homogeneity of the eluted fractions was determined via Coomassie Brilliant Blue stained SDS-PAGE. Samples were concentrated in vivaspin columns (Sartorius).

## Kinase reactions

Abl kinase treated CTD reactions were prepared in buffer conditions containing 1 µg/µL GST-yCTD substrate, 0.0035 µg/µL c-Abl kinase, 50 mM Tris-HCl at pH7.5, 50 mM MgCl$_2$ and 2 mM ATP. TFIIH treated CTD reaction were prepared in buffer conditions containing 1 µg/µL GST-CTD substrate, 0.025 µg/µL TFIIH, 50 mM Tris-HCl at pH7.5, 50 mM MgCl$_2$ and 2 mM ATP. P-TEFb treated CTD reaction, were prepared in buffer conditions containing 1 µg/µL GST-CTD substrate, 0.0075 µg/µL P-TEFb, 50 mM Tris-HCl at pH7.5, 50 mM MgCl$_2$ and 2 mM ATP. Erk2-treated CTD reaction, as well as the controls with no kinase treatment, were prepared in buffer conditions containing 1 µg/µL GST-CTD substrate, 0.025 µg/µL Erk2, 50 mM Tris-HCl at pH7.5, 50 mM MgCl$_2$ and 2 mM ATP. These reactions were incubated for various amount of time at 30℃ along with control experiments setup under identical conditions but without kinases and then stored at −80℃ until analysis.

Tandem kinase treatments were performed by mixing 10 µg GST-CTD substrate treated with the c-Abl as described above (incubated overnight for 16 hr) and an equal volume of a solution containing the second kinase (0.05 µg/µL TFIIH or 0.015 µg/µL P-TEFb or 0.05 µg/µL Erk2) in tandem reaction buffer (50 mM Tris-HCl at pH7.5, 50 mM MgCl$_2$, 2 mM ATP). These were incubated at 30℃ for 16 hr and stored at −80℃ until analysis.

## Kinase activity assay

The Erk2 kinetic activity assay was performed in a 25 µl reaction volume containing 0–100 µM substrate (GST yCTD or GST YtoF CTD) and a reaction buffer of 40 mM Tris-HCl at pH 8.0 and 20 mM MgCl$_2$. The reaction was initiated by adding 187 nM of Erk2 and incubated at 28℃ for 15mins before being quenched with 25 µl H$_2$O and 50 µl of room temperature Kinase-Glo Detection Reaction (Promega). The mixtures were allowed to sit at room temperature for 10 min before reading the bioluminescence in a Tecan Plate reader 200. The readings obtained were translated to ATP concentration with the help of an ATP standard curve determined with the Kinase Detection Reagent.

The TFIIH kinetic reactions were set up with 0–100 µM substrate (GST yCTD or GST YtoF CTD), 0.2 µM TFIIH, 0.1 mg/ml Bovine Serum Albumin (BSA) and reaction buffer of 50 mM Tris-HCl pH 8.0, 10 mM MgCl$_2$, 1 mM DTT. 500 µM ATP Mix (10 nCi/µl radiolabeled ATP, PerkinElmer) was added to each tube to start the reactions. The tubes were subsequently incubated in a 30℃ water bath for 30mins and quenched with 500 µl of quench buffer (1 mM potassium phosphate pH 6.8, 1 mM EDTA) to a reaction volume of 10 µl. Each reaction was loaded onto 0.45 µm nitrocellulose filters and washed three times with 1 mM potassium phosphate buffer to remove any excess labeled ATP. Filters were added to glass vials with scintillation fluid, Econo-Safe Economical Biodegradable Counting Cocktail (Research Products International) and set in a scintillation counter for 5 min reads each. The amount of phosphate incorporation was determined for each reaction using a set of 147 pmol labeled ATP standards that were read alongside each reaction set.

Kinetic data obtained from the two assays described above were analyzed in R (*Hamilton, 2015*; *R Development Core Team, 2017*) and fitted to the Michaelis-Menten kinetic equation to obtain respective kinetic parameters $k_{cat}$ (s$^{-1}$) and $K_m$ (µM).

## Electrophoretic mobility shift assay

SDS-PAGE analysis was performed using 10–15% acrylamide gels containing 1% SDS. GST-CTD samples were prepared by boiling with SDS-PAGE loading dye at 95℃ for 5 min. This was also used to quench time-course reactions. A volume containing approximately 1 µg of phosphorylated GST-CTD substrate or no kinase control was loaded into wells and resolved at ~150V for 1 hr at room temperature. Gels were stained with Coomassie Brilliant Blue and visualized on G: BOX imaging systems (Syngene).

## MALDI-MS analysis

Approximately 5 µg of GST-CTD protein from the kinase reactions described were prepared for MALDI-MS. If necessary, the protein was digested with 3C protease by mixing sample in a 1:10 ratio of 3C-protease to GST-CTD variants. Proteins were equilibrated with dilute trifluoracetic acid (TFA) to a final concentration of 0.1% TFA and a pH of <4. These samples were desalted using ZipTip (Millipore) tips according to manufacturer's instructions. These samples were mixed 1:1 with a 2,5-dihydroxybenzoic acid matrix solution (DHB) and spotted on a stainless steel sample plate. The spots

were allowed to crystallize at ambient temperature and pressure. MALDI-MS spectra were obtained on an AB Voyager-DE PRO MALDI-TOF instrument with manual adjustment of instrument parameters to ensure the greatest signal to noise. Sample masses were determined by an internal calibration against the untreated GST-CTD variants. Data analysis, noise reduction, and Gaussian smoothing, if necessary, were performed in DataExplorer (AB), R, and the R package smoother (*Hamilton, 2015*; *R Development Core Team, 2017*). Masses were determined as the highest local intensity peak of the post-processed data. Data were visualized in R-Studio using ggplot2 (*Wickhan, 2009*). All MALDI experiments were carried out three times independently with biological triplicates.

## Cell culture and total protein preparation

HEK293T cells were purchased from ATCC and no mycoplasma contamination was detected. The cells were maintained in DMEM (ISC BioExpress cat#T-2989–6) with splitting every other day and seeding at a concentration of $9.6 \times 10^4$ cells per 10 cm culture dish and incubated at 37°C at 5% $CO_2$. Cells to be treated with imatinib, dasatinib, or vehicle control were plated at $5 \times 10^5$ cells per well in 6-well tissue culture plates in fresh DMEM. Cells were incubated for 24 hr, and the media was replaced with fresh DMEM containing the indicated amount of inhibitor or equivalent portion of DMSO vehicle control for an additional 24–40 hr.

Protein preparations were generated by direct in-well lysis. Media was then removed, and the cells were washed with ice-cold Phosphate-buffered saline (PBS), and 200 μL RIPA buffer (150 mM NaCl, 10 mM Tris-HCl pH 7.5, 0.1% SDS, 1% Triton X-100, 1% deoxycholate, and 5 mM EDTA) supplemented to 1X with HALT protease and phosphatase inhibitor cocktail (Thermo Scientific) was added directly to cells. Plates were incubated on ice for 15 min with gentle shaking, and the lysate was transferred to microcentrifuge tubes. Samples were briefly sonicated to reduce viscosity and spun to remove cell debris. Protein concentration was determined to utilize Pierce BCA Protein Assay Kit (Thermo Scientific) against a BSA standard curve. Samples were diluted with SDS-PAGE loading buffer and boiled at 95°C for 5 min. The sample was aliquoted and frozen at −80°C.

## Immunoblotting

Total protein from cell lysate (20–40 μg) or GST-yCTD samples (50 to 500 ng, dependent on epitope) was loaded onto a 4–20% gradient SDS-PAGE gel (Biorad, Cat#:456–1096) and ran at 150V for 50 min at room temperature in a Mini-PROTEAN Tetra Cell (Biorad). The proteins were transferred to PVDF membrane at 100 V for 1 hr on ice in a Mini-PROTEAN Tetra Cell (Biorad). Membranes were blocked in 1X TBST (20 mM Tris-HCl pH 7.5, 150 mM NaCl, 0.1% Tween-20) and 5% (BSA) or non-fat dry milk for 1 hr at room temperature with shaking. Blocked membranes were incubated in primary antibodies in either 1X TBST or 1X TBST+5% BSA at 4°C overnight or 1 hr at room temperature. The membranes were then washed six times with 1X TBST for 5 min each at room temperature and incubated with secondary antibodies in 1X TBST for 1 hr at room temperature. The membrane was washed once again and incubated with SuperSignal West Pico Chemiluminescent Substrate (Pierce) according to factory directions. Blots were imaged using a G:BOX gel doc system (Syngene) and quantified in ImageJ (*Schneider et al., 2012*). Statistical analysis was performed in R (*R Development Core Team, 2017*).

Blots normalized against Coomassie-stained bands were stained by incubating the membrane post-immunoblotting with stain solution (0.1% Coomassie Brilliant Blue R-250, 40% ethanol, 10% acetic acid) for 1 min. The stain was discarded, and the blot was briefly rinsed with distilled water. Blot was de-stained in de-stain solution (10% ethanol, 7.5% acetic acid) until bands were visible. Blots were equilibrated with distilled water and imaged wet in a plastic blot protector using a G:BOX gel doc system (Syngene) and quantified as above.

For dot blot, samples of GST-yCTD were treated with P-TEFb alone or c-Abl followed by P-TEFb. The heat-inactivated samples were prepared by heating the c-Abl treated GST-yCTD at 60°C for five minutes, while the Dasatinib inactivated samples were prepared by adding 10 μM Dasatinib to the c-Abl treated sample 15 min before incubation with P-TEFb. The dot blots were performed by adding 2X SDS page loading dye and briefly heating each sample after which 1 μg of the sample was loaded in three replicates onto a 0.45 μm Nitrocellulose membrane. The membrane was subsequently allowed to dry and blocked in 1X TBST (20 mM Tris-HCl pH 7.5, 150 mM NaCl, 0.1% Tween-

20) and 5% BSA for 1 hr at room temperature with shaking. The Anti-RNA Pol II phosphoSer2 antibody, 3E10, (Millipore) was diluted 1:5000 times and was incubated overnight at 6°C to probe for Ser2 phosphorylation. The membranes were then washed five times with 1X TBST for 5 min each at room temperature and incubated with secondary antibody in 1X TBST for 1 hr at room temperature. The membrane was washed once again and incubated with SuperSignal West Pico Chemiluminescent Substrate (Pierce) according to manufacturer's instruction. Blots were imaged using a G:BOX gel doc system (Syngene) and quantified in ImageJ (*Schneider et al., 2012*). Statistical analysis was performed by using the Data Analysis function in Microsoft Excel.

Primary and secondary antibodies were stripped for re-probing by incubating membranes with a mild stripping buffer (200mM glycine, 0.1% SDS, 1% Tween 20, pH 2.2) for 10 minutes. Stripping buffer was discarded and this step was repeated once more. The blot was washed twice with 1X PBS for 10 minutes. The blot was washed twice with 1X TBST for 10 minutes. The membrane was blocked and re-probed with secondary antibody and chemiluminsecent reagent, as described above, to insure complete removal of both primary and secondary antibodies. The membrane was then probed for desired epitope as described above.

## LC-UVPD-MS analysis

GST-3CTD samples (approximately 1 μg/μL) were digested on ice for 4 hr in 50 mM Tris-HCl at pH 8.0 with 150 mM NaCl using 3C-protease at a molar ratio of 100:1 protein: protease in a reaction volume of 20 μL. Digests were desalted on C18 spin columns and resuspended to 1 μM with 0.1% formic acid for LC-MS analysis.

Separations were carried out on a Dionex Ultimate 3000 nano liquid chromatograph plumbed for direct injection. Picofrit 75 μm id analytical columns (New Objective, Woburn, MA) were packed to 20 cm using 1.8 μm Waters Xbridge BEH C18 (Milford, MA). Mobile phase A was water, and B was acetonitrile, each containing 0.1% formic acid. Separations occurred over a 30 min linear gradient from 2–35% B. The flow rate was maintained at 0.3 μL/min during the separation.

An Orbitrap Fusion Lumos Tribrid mass spectrometer (Thermo Fischer Scientific, San Jose, CA) equipped with a Coherent ExciStar XS excimer laser operated at 193 nm was used for positive mode LC-MS/MS analysis of the 3CTD peptides. The Lumos mass spectrometer was modified for ultraviolet photodissociation (UVPD) as described earlier (*Klein et al., 2016*). Photoactivation in the low-pressure linear ion trap was achieved using 2 pulses at 2 mJ in a targeted *m/z* mode. The 3+ charge states of the singly and doubly phosphorylated peptide GPGSGMYSPTSPSYSPTSPSYSPTSPS were targeted for photoactivation. All data were acquired in the Orbitrap analyzer where MS1 and MS/MS spectra were collected at resolving powers of 60K and 15K (at *m/z* 200), respectively. MS1 spectra were acquired from *m/z* 400–2000 with an AGC setting of 5E5. Each MS/MS spectrum consisted of two microscans collected from *m/z* 220–2000 with an AGC setting of 2E5.

Data analysis was performed using the XCalibur Qual Browser and ProSight Lite (*Fellers et al., 2015*). For both targeted *m/z* values, the MS/MS spectrum for each phosphoform present was deconvoluted to neutral forms using Xtract with a signal-to-noise threshold of 3. Sequence coverage was determined by matching the nine ion types observed with UVPD (a, a•, b, c, x, x•, y, y-1, z). Localization of the phosphorylation(s) was performed by adding a phosphate group (+79.966 Da) at each of the possible serine, threonine, and tyrosine residues to identify fragment ions containing the moiety and optimize characterization scores in ProSight Lite.

Analysis of yCTD treated with P-TEFb was performed identically to previous analysis of yCTD treated with TFIIH and Erk2 (*Mayfield et al., 2017*). GST-yCTD samples were prepared for bottom-up analysis using a two-step proteolysis method. First, overnight digestion with trypsin at 37°C was carried out using a 1:50 enzyme to substrate ratio, which cleaved the GST-portion of the protein while leaving the abasic 26mer CTD portion intact. The resulting digest was filtered through a 10 kDa molecular weight cutoff (MWCO) filter to remove tryptic GST peptides and buffer exchange the CTD portion into 50 mM Tris-HCl pH 8.0 and 10 mM CaCl₂ for subsequent proteinase K digestion. Proteinase K was added in a 1:100 ration and digested overnight at 37°C. Samples were diluted to 1 μM in 0.2% formic acid for LC-MS. Analysis of yCTD-Lys treated by c-Abl, P-TEFb or c-Abl followed by P-TEFb is using a similar method as described above except the first digestion was done by 3C-protease and second by trypsin.

A bottom-up analysis of yCTD was performed on a Velos Pro dual linear ion trap mass spectrometer (Thermo Fisher) equipped with a Coherent ExciStar XS excimer laser (Santa Clara) at 193 nm and

500 Hz as previously described for UVPD (*Gardner et al., 2008*; *Madsen et al., 2010*). Two pulses of 2mJ were used for photodissociation. Separations were carried out on a Dionex Ultimate 3000 nano liquid chromatography (Thermo Fischer) configured for preconcentration. Integrafrit trap columns were packed to 3.5 cm using 5 µm Michrom Magic C18. Picofrit analytical columns were packed to 20 cm using 3.5 µm Waters Xbridge BEH C18 (Waters). Mobile phase A was water, and mobile phase B was acetonitrile; each contained 0.1% formic acid. Peptides were loaded onto the trap column for 5 min in an aqueous solvent containing 2% acetonitrile and 0.1% formic acid at a 5 µL/min flow rate. Separations occurred over a 20 min linear gradient in which percent phase B was increased from 2–15% during the first 15 min and further increased to 35% over the last 5 min. The flow rate was constant at 0.3 µL/min. A top seven data-dependent acquisition method was first used to identify the main phosphorylated species. A targeted analysis followed in which the singly phosphorylated heptad peptides were continually selected for UVPD activation (between MS$^1$ acquisitions occurring after every five MS/MS events) to resolve partially co-eluting phospho-isomers. All MS experiments were carried out three times independently with biological triplicates. Resulting UVPD spectra were manually interpreted.

## ChIP-seq analysis

For ChIP-seq experiments, HEK293T cells were seeded at 3.5 million cells in a 15 cm dish. After 24 hr, when the cells achieved a confluence of 40–50%, the media was replaced by fresh media containing 10 µM Dasatinib inhibitor or the DMSO control and allowed to grow for another 24 hr until the confluence of 80% was achieved. The cells were fixed with 1% formaldehyde in 15 ml of media, for 8 min at room temperature with intermittent swirling. The reaction was quenched by the addition of glycine to a final concentration of 0.125M and incubation for five minutes at room temperature. The cells were washed twice with 15 ml of ice-cold Dulbecco's phosphate-buffered saline and scraped off the surface. The cells were pelleted at a speed of 8000 g for 5 min, resuspended and aliquoted such that the number of control cells (with only DMSO) were normalized to the number of dasatinib treated cells. The cell pellet was frozen in a freezing mixture comprised of dry ice and ethanol.

The cells were lysed by adding buffer LB1 [50 mM HEPES at pH 7.5, 140 mM NaCl, 1 mM EDTA, 10% glycerol, 0.5% NP-40, 0.25% Triton-X 100, 1x Protease inhibitor cocktail (Thermoscientific)] and placing the tubes on a rotating wheel at 4°C for 10 min, following which they were spun at 2000 g for 5 min to isolate the nuclei as a pellet. These were washed with buffer LB2 [10 mM Tris-HCL at pH 8.0, 200 mM NaCl, 1 mM EDTA, 0.5 mM EGTA + 1X Protease inhibitor cocktail (thermoscientific)], and subsequently the nuclei where resuspended in 300 µl of nuclear lysis buffer LB3 [10 mM Tris-HCL, pH 8, 100mMNaCl, 1 mM EDTA, 0.5 mM EGTA, 0.1% Na-Deoxycholate, 0.5% N-lauroylsarcosine and 1x Protease inhibitor cocktail (Thermoscientific)].

The nuclear lysate of 300 µl was sonicated using a Biorupter UCD 200 (Diagenode) for 25 cycles at maximum intensity (15 s ON 45 s OFF in a water bath at 4°C). After each of the 10 cycles, the samples were incubated on ice for 10 min. Following sonication 30 µl of buffer LB3 supplemented with 10% Triton X-100 was added into the sample and spun at full speed for 10 min to remove cell debris. 30 µl of the supernatant was taken as the input control for ChIP-seq, and the rest is used to prepare the samples.

Magnetic Protein-G beads (Thermo Fischer) were incubated with respective antibody (1 µg per 10 µl of beads) overnight on the rotating shaker at 4°C. The beads were then washed thrice with 5% BSA in PBS to remove any excess antibody, and the 300 µl of the sonicated lysate prepared above is added to it, with 800 µl of buffer LB3 and 100 µl of buffer LB3 supplemented with10% Triton X-100. The samples were placed on a rotating wheel overnight at 4°C for the immunoprecipitation to occur. The beads were washed twice by a low salt buffer (0.1% sodium deoxycholate, 1% Triton X-100, 1 mM EDTA, 50 mM HEPES at pH 7.5, 150 mM NaCl) followed by wash with high salt buffer (0.1% Na Deoxycholate, 1% Triton X-100, 1 mM EDTA, 50 mM HEPES at pH 7.5, 500 mM NaCl), lithium chloride buffer (250 mM LiCl, 0.5% NP-40, 0.5% Na Deoxycholate, 1 mM EDTA, 10 mM Tris-HCl at pH 8.1) and finally washed twice with TE buffer (10 mM Tris-HCl at pH 8.1 and 1 mM EDTA). The beads were ultimately resuspended in 200 µl of elution buffer (1% SDS and 0.1M sodium bicarbonate) and placed in the thermomixer at 65°C for 16 hr to enable reverse crosslinking.

Both the input and treatment samples were with 70 µl of elution buffer (1% SDS and 0.1M sodium bicarbonate) and underwent reverse crosslinking at 65°C for 16 hr. After the reverse crosslinking, phenol-chloroform extraction was used to extract the immunoprecipitated DNA, Library prep was

done using a starting amount of 3 ng of DNA measured by Qubit HS (Thermo Fischer) using the NEBNext Ultra II DNA Library Prep Kit for Illumina (NEB) following the vendor manual. The libraries with multiplex index primers prepared above were pooled together and sequenced using the Next-Seq single end 75 base pair sequencing platform.

Reads were aligned to the human genome (hg19) using bowtie with '–best –strata –m 1' parameters (*Langmead et al., 2009*). Only uniquely mapped reads were selected for downstream analysis. MACS2 was employed to call peaks by comparing immunoprecipitated chromatin with input chromatin using standard parameters and a q-value cutoff of 1e-5 (*Zhang et al., 2008*). The peaks overlapped with the blacklist regions downloaded from UCSC were removed. Each sample was normalized to 10 million mapped reads and visualized in Integrative Genomics Viewer (IGV) (*Robinson et al., 2011*). The pausing index was defined as the ratio of Pol II density in the promoter-proximal region and the Pol II density in the transcribed region (*Zeitlinger et al., 2007*). The proximal promoter region is defined as −50 bp to +300 bp around the transcription start site (TSS); while the transcribed region (gene body) is from +300 bp to the 3000 bp downstream of transcription end site (TES) (*Rahl et al., 2010*).

## Quantification and data analysis

MALDI data analysis, noise reduction, and Gaussian smoothing (if necessary) were performed in DataExplorer (AB), R, and the R package smoother to provide interpretable data (*Hamilton, 2015*; *R Development Core Team, 2017*). Data were visualized in R-Studio using ggplot2 (*Wickhan, 2009*). LC masses were determined as the highest local intensity peak of the post-processed data. Tandem mass spectrometry data analysis was performed using the XCalibur Qual Browser and ProSight Lite. For both targeted $m/z$ values, the MS/MS spectrum for each phosphoform present was deconvoluted to neutral forms using Xtract with a signal-to-noise threshold of 3. Sequence coverage was determined by matching the nine ion types observed with UVPD ($a$, $a^{\bullet}$, $b$, $c$, $x$, $x^{\bullet}$, $y$, $y$-1, $z$). Localization of the phosphorylation(s) was performed by adding a phosphate group (+79.966 Da) at each of the possible serine, threonine, and tyrosine residues to identify fragment ions containing the moiety and optimize characterization scores in ProSight Lite (*Fellers et al., 2015*). Western blots were quantified using ImageJ (*Schneider et al., 2012*) and statistical significance was determined by two-tailed unpaired Student's t-test assuming unequal variances in Microsoft Excel. Statistical significance is reported in the figure legends. Results have been shown with ± standard deviation or SEM, as mentioned in the figure legends.

Access Code: The ChIP-seq data for RNA polymerase II have been deposited into GEO with access codes GSE131838.

## Acknowledgements

We want to thank Drs. G Gill and J Dixon for comments on the manuscript. This work is supported by grants from the National Institutes of Health (R01 GM104896 and 125882 to YJZ and R21EB018391 to JSB) and Welch Foundation (F-1778 to YJZ and F-1155 to JSB). Voyager DE-PROT MALDI MS data were collected in the University of Texas at Austin Proteomics Facility. Funding from the UT System for support of the UT System Proteomics Core Facility Network is gratefully acknowledged.

## Additional information

### Funding

| Funder | Grant reference number | Author |
| --- | --- | --- |
| National Institute of General Medical Sciences | R01 GM104896 | Yan Zhang |
| National Institute of General Medical Sciences | R01 RM125882 | Yan Zhang |
| National Institute of Biomedical Imaging and Bioengineering | R21EB018391 | Jennifer Brodbelt |

| Welch Foundation | F-1778 | Yan Zhang |
| Welch Foundation | F-1155 | Jennifer Brodbelt |

The funders had no role in study design, data collection and interpretation, or the decision to submit the work for publication.

### Author contributions
Joshua E Mayfield, Conceptualization, Data curation, Formal analysis, Investigation, Writing—original draft; Seema Irani, Data curation, Formal analysis, Validation, Writing—review and editing; Edwin E Escobar, Data curation, Formal analysis, Writing—original draft; Zhao Zhang, Formal analysis, Visualization; Nathaniel T Burkholder, M Rachel Mehaffey, Sarah N Sipe, Wanjie Yang, Data curation, Validation; Michelle R Robinson, Data curation, Methodology; Nicholas A Prescott, Data curation, Preparation of the enzymes used in the experiments; Karan R Kathuria, Data curation, Preparation of the DNA constructs used in the experiments; Zhijie Liu, Formal analysis, Writing—review and editing; Jennifer S Brodbelt, Formal analysis, Validation, Writing—review and editing; Yan Zhang, Conceptualization, Data curation, Formal analysis, Supervision, Funding acquisition, Validation, Visualization, Writing—original draft, Project administration, Writing—review and editing

### Author ORCIDs
Seema Irani (iD) https://orcid.org/0000-0001-5159-3473
Edwin E Escobar (iD) http://orcid.org/0000-0002-0086-6264
Sarah N Sipe (iD) http://orcid.org/0000-0003-4554-7571
Nicholas A Prescott (iD) https://orcid.org/0000-0002-0635-8906
Jennifer S Brodbelt (iD) https://orcid.org/0000-0003-3207-0217
Yan Zhang (iD) https://orcid.org/0000-0002-9360-5388

### Decision letter and Author response
Decision letter https://doi.org/10.7554/eLife.48725.019
Author response https://doi.org/10.7554/eLife.48725.020

## Additional files

### Supplementary files
• Transparent reporting form
DOI: https://doi.org/10.7554/eLife.48725.015

### Data availability
All mass spec data generated or analyzed during this study are included in the manuscript and supporting files. Source data files have been provided in Figure 3F, Figure 1—figure supplement 2 and Figure 3—figure supplement 2 and 4.

The following dataset was generated:

| Author(s) | Year | Dataset title | Dataset URL | Database and Identifier |
|---|---|---|---|---|
| Irani S, Zhang Z, Liu Z, Zhang Y | 2019 | RNA polymerase II ChIP with dasatinib inhibition | https://www.ncbi.nlm.nih.gov/geo/query/acc.cgi?acc=GSE131838 | NCBI Gene Expression Omnibus, GSE131838 |

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
