## [Decision Letter]

Thank you for submitting your article "Tyr1 phosphorylation promotes the phosphorylation of Ser2 on the C-terminal domain of RNA polymerase II by P-TEFb" for consideration by *eLife*. Your article has been reviewed by three peer reviewers, including Jerry L Workman as the Reviewing Editor and Reviewer #1, and the evaluation has been overseen by Philip Cole as the Senior Editor. The following individual involved in review of your submission has agreed to reveal their identity: Nicolas Young (Reviewer #2).

The reviewers have discussed the reviews with one another and the Reviewing Editor has drafted this decision to help you prepare a revised submission.

Summary:

In this manuscript, authors showed the importance of the pol II CTD Tyr1 residue and Tyr1 phosphorylation (pTyr1) on CTD serine kinases, especially P-TEFb, specificity regulation by using in vitro kinase assays and Mass spec analysis. They showed that Tyr1 mutation into phenylalanine alters the CTD phosphorylation pattern by Erk2, TFIIH, and P-TEFb kinases. By in vitro phosphorylation of CTD construct by c-Abl kinase, they could create the CTD constructs whose Tyr1 residues are partially phosphorylated. On partial pTyr1 containing CTD, P-TEFb showed increased specificity toward CTD Ser2, rather than Ser5, which is the preferred target when CTD is not phosphorylated by c-Abl. Authors also provide in vivo evidence supporting their in vitro experiments. pTyr1 reduction by treatment with c-Abl specific inhibitor imatinib, or Abl like kinase inhibitor dasatinib results in CTD Ser2 phosphorylation (pSer2) reduction while Ser5 phosphorylation (pSer5) remains unchanged. In addition, dasatinib treatment causes increased Pol II pausing index and reduced CTD pSer2 at genes having a high pausing index.

This study provides an attractive model for understanding the discrepancy between in vivo and in vitro specificity differences of P-TEFb against Pol II CTD. Their in vitro assays combined with mass spec analysis are well designed. Although the in vivo phenotypes are not clear enough to prove their model, this may be due to the redundant nature of CTD Tyr1 kinases in vivo, and the shortage of tools to sufficiently and selectively reduce CTD pTyr1 signal.

Essential revisions:

1) In their in vitro kinase assays, the authors repeatedly used the term 'saturating conditions' which is confusing. They did not provide any explanation or data supporting that their experimental conditions are really saturating. Especially in Figure 2, as partial phosphorylation of CTD by c-Abl1 is very important for their overall logic, they need to provide a better explanation or data showing that this observation is not caused by insufficient incubation or kinase amount. They need to more precisely define their terms, or just more accurately describe their experimental conditions.

2) This work hypothesizes that CTD pTyr1 precedes pSer2. But CTD pTyr1 can be done at CTD containing pSer2 or pSer5. Does pre-existing CTD pSer2 or pSer5 affect c-Abl activity against pTyr1?

3) Although CTD pTyr1 in vivo seems to be achieved by multiple kinases including c-Abl, this work shows that c-Abl specific Imatinib treatment is enough to cause pTyr1 and pSer2 reduction in Figure 2C and Figure 4B. Is CTD pTyr1 and pSer2 reduced upon C-Abl knock down?

4) Imatinib is a highly selective C-Abl kinase inhibitor whereas dasatinib has broader specificity against abl like tyrosine kinases. However, as a control, the authors need to show or refer to evidence that those two kinase inhibitor molecules do not affect P-TEFb activity, at least in vitro.

5) In Figure 3—figure supplement 1, the authors insist that they kill the activity of c-Abl by heat inactivation or by treatment with dasatinib. However, the authors did not show the proof that c-Abl activity is sufficiently blocked by these treatments. They need to show their titration data using increasing amount of dasatinib or increasing heat inactivation time to fully kill the c-Abl activity against CTD Tyr1.

6) The authors fail to specifically identify the protein that they are studying, only referring to it obliquely as the CTD of RNAP II. While this is an obvious reference to the majority of the audience, it would be better to at some point specifically mention the protein (BPB1) and not only the complex.

7) Given the excellent work, it would be appropriate to also point out what has not been done and what future directions could be fruitful. It was particularly noticed that the actual patterns of phosphorylation between heptads (inter-heptad) was not resolved or well-studied here as technical challenges remain.

8) Figure 4: There is a reference to Supplementary Figure 13 but we did not find this figure. We were concerned that the interplay found here might result from technical crosstalk through modulation of antibody response via neighboring PTMs. This concern appears to be sufficiently addressed by the MS data. It may be helpful to the reader to more specifically declare that this may be a problem; however, the MS data appears to support the Western blot data.

As the main claim made by authors is that Y1P alters p-TEFb's specificity from Ser5 to Ser2, several key details need to be addressed more rigorously:

9) If c-Abl-phosphorylated CTD causes elevated Ser2P levels by p-TEFb while Ser5P remains the same (Figure 3B and C), why did the overall phosphorylation level of tandem reactions decreases comparing to p-TEFb acting alone (24-29 to 16-26)?

10) Can the authors quantify the overall levels of Ser2P and Ser5P when c-Abl-pretreated K7-CTD was used? Is Ser5P still the predominant form of products by p-TEFb? This would be an important parameter to conclude whether Y1P merely promotes Ser2P vs. Y1P alters Ser5P to Ser2P.

11) Does p-TEFb cause further gel-shift of c-Abl-pretreated CTD?

12) Descostes et al., 2014, showed that Tyr1P co-immunoprecipitated with Ser5P and Ser7P but not Ser2P or Thr4P. Can the authors reconcile this seeming discrepancy between the above observations and their main conclusion?

13) The authors should consider re-purifying the phosphorylated CTD after the c-Abl reaction so as to completely rule out the possibility that kinase-kinase interaction might cause the p-TEFb activity switch.

14) It wasn't clear to this reviewer why dmCTDs (WT and YtoF mutant) were used in part of Figure 1, then authors changed to yeast CTD for the rest of kinetics experiments. If similar results were obtained from both yeast and fly, it might be easier to show one complete set of data for consistency.

15) The author should consider stressing that continuous Y1P inhibits p-TEFb-mediated phosphorylation (Czudnochowski, 2012) whereas c-Abl-pretreated CTD displays an alternate Y1P pattern which promotes Ser2P phosphorylation.

---

## [Author Response]

Essential revisions:

*1) In their* in vitro *kinase assays, the authors repeatedly used the term 'saturating conditions' which is confusing. They did not provide any explanation or data supporting that their experimental conditions are really saturating. Especially in Figure 2, as partial phosphorylation of CTD by c-Abl1 is very important for their overall logic, they need to provide a better explanation or data showing that this observation is not caused by insufficient incubation or kinase amount. They need to more precisely define their terms, or just more accurately describe their experimental conditions.*

This is a wonderful point. We considered a reaction condition saturating when we don’t detect further addition of phosphate to the CTD substrate with longer incubation or addition of kinase/ATP using MALDI-TOF. In the revision, we have replaced the word “saturating” with the specific conditions of the reactions and explained further incubation doesn’t increase phosphorylation.

2) This work hypothesizes that CTD pTyr1 precedes pSer2. But CTD pTyr1 can be done at CTD containing pSer2 or pSer5. Does pre-existing CTD pSer2 or pSer5 affect c-Abl activity against pTyr1?

This is a great question. We observed in MALDI-TOF that when every Ser5 of the CTD is phosphorylated, c-Abl cannot place more phosphorylation at Tyr1. In regards to pSer2’s influence, ChIP-seq analysis suggests pTyr1 occurs prior to pSer2 accumulated only after pTyr1 and, therefore, the reverse of this order may not be physiologically relevant.

3) Although CTD pTyr1 in vivo seems to be achieved by multiple kinases including c-Abl, this work shows that c-Abl specific Imatinib treatment is enough to cause pTyr1 and pSer2 reduction in Figure 2C and Figure 4B. Is CTD pTyr1 and pSer2 reduced upon C-Abl knock down?

Indeed, it was shown that ABL2 could also phosphorylate RNA polymerase II at Tyr1 when c-Abl is abolished (Baskaran et al., 1997). Our approach to inhibit c-Abl with small molecules such as imatinib inhibits not only c-Abl but also ABL2. Because of the high sequence conservation between the kinase domains of the two enzymes (93%), imatinib inhibits ABL2 with a *Ki* around 3nM and binds ABL2 in a highly similar manner to ABL1 according to the co-crystal structures (PDB code 3GVU, Salah E et al., 2011). We were able to see a reduction of pTyr1 and pSer2 since the inhibitor blocks both kinases’ function. On the other hand, ABL1 knockdown only reduces c-Abl, and this loss can be compensated for by ABL2 making it difficult to observe a drop in the global amount of Tyr1 phosphorylation using western blot of whole-cell lysate. Our results are consistent with other reported results showing that global pTyr1 levels were unaffected by shRNA against ABL1, even though they observed alterations in pTyr1-linked phenomena investigated through other means (Burger et al., 2019). Therefore, ABL1 likely contributes to a portion of global Tyr1 phosphorylation but is unlikely to account for all of it.

*4) Imatinib is a highly selective C-Abl kinase inhibitor whereas dasatinib has broader specificity against abl like tyrosine kinases. However, as a control, the authors need to show or refer to evidence that those two kinase inhibitor molecules do not affect P-TEFb activity, at least* in vitro.

Following the reviewers’ advice, we have monitored P-TEFb activity in the presence of c-Abl inhibitors. In this assay, we incubated P-TEFb with substrate CTD and varying amounts of dasatinib, the compound used in our biochemical assays and ChIP-seq samples. Using EMSA, we show that P-TEFb still adds phosphate to the CTD at concentrations as high as 20µM, but c-Abl shows no activity at dasatinib concentrations beyond 2.5µM. Similar results were also observed using imatinib. These results have been added as Figure 3—figure supplement 1C to show that c-Abl inhibitors don’t cross-inhibit P-TEFb.

5) In Figure 3—figure supplement 1, the authors insist that they kill the activity of c-Abl by heat inactivation or by treatment with dasatinib. However, the authors did not show the proof that c-Abl activity is sufficiently blocked by these treatments. They need to show their titration data using increasing amount of dasatinib or increasing heat inactivation time to fully kill the c-Abl activity against CTD Tyr1.

We followed the reviewers’ suggestion and added a supplementary figure where we show the dose-response of c-Abl heat inactivation (Figure 3—figure supplement 1B). As shown in the EMSA assay, c-Abl has little kinase activity after 2 minutes incubation at 65°C.

6) The authors fail to specifically identify the protein that they are studying, only referring to it obliquely as the CTD of RNAP II. While this is an obvious reference to the majority of the audience, it would be better to at some point specifically mention the protein (BPB1) and not only the complex.

This is a good point. We have substituted RPB1 in appropriate places in the revision.

7) Given the excellent work, it would be appropriate to also point out what has not been done and what future directions could be fruitful. It was particularly noticed that the actual patterns of phosphorylation between heptads (inter-heptad) was not resolved or well-studied here as technical challenges remain.

Thank you for the excellent suggestion to improve our manuscript. We have added a future direction section to the end of our Discussion. Specifically, we have included the following text “Future inquiries using similar multi-disciplinary approaches will hopefully reveal CTD modification patterns in greater detail, across a greater number of contiguous heptads, and at different stages of the transcription process in single amino acid resolution. Information such as this will define the temporal and spatial signaling allowing for the recruitment of transcriptional regulators during active transcription.”

8) Figure 4: There is a reference to Supplementary Figure 13 but we did not find this figure. We were concerned that the interplay found here might result from technical crosstalk through modulation of antibody response via neighboring PTMs. This concern appears to be sufficiently addressed by the MS data. It may be helpful to the reader to more specifically declare that this may be a problem; however, the MS data appears to support the Western blot data.

We thank the reviewers for catching this mis-numbering. We have corrected the error, and this section now refers appropriately to Figure 4—figure supplement 1. The interplay of antibody recognition is indeed a major concern, which was our motivation to develop mass spectrometry methodologies to identify phosphorylated species. We are adding discussion of the potential issue of cross-talk to the Results section alongside our interpretation when compounded with the mass spectrometry data.

As the main claim made by authors is that Y1P alters p-TEFb's specificity from Ser5 to Ser2, several key details need to be addressed more rigorously:9) If c-Abl-phosphorylated CTD causes elevated Ser2P levels by p-TEFb while Ser5P remains the same (Figure 3B and C), why did the overall phosphorylation level of tandem reactions decreases comparing to p-TEFb acting alone (24-29 to 16-26)?

We thank the reviewers for making this observation and encouraging us to clarify. We easily captured the pronounced increase in Ser2 phosphorylation because it is increasing from very low abundance (Figure 1A and 3B) to the primary species when flanking heptads have pTyr1 (Figure 3G). We only detect a small and statistically non-significant decrease in Ser5 using western blot (Figure 3B). However, we cannot conclude from this blot that pSer5 remains at the same level. First, the products are a mixture containing tandem treatment product as well as a significant amount of CTD substrate that didn’t react with c-Abl in the first step. If tandem products only account for a small fraction of overall material (which can happen in in vitro biochemical assays), their quantity changes might not be detected with statistical significance. Thus, we cannot conclude if pSer5 for the tandem reaction is altered or not because the mixtures are not separately quantified. Furthermore, the antibody recognition used in the western blot might affect the reliability of the quantification. This motivated us to analyze the product mixture using LC-UVPD-MS analysis where different products can be isolated and directly detected. We were able to isolate the pTyr1 containing di-peptides, which are the products of tandem treatment. Upon isolation, pSer2 is much more enriched than pSer5 in the peptide that also contains pTyr1 (Figure 3G peak 7 and 8 vs. 6 and 9). We have added in the revision the reason driving us to study the tandem reaction products with LC-UVPD-MS.

The apparent reduction in the number of phosphates added by P-TEFb in c-Abl treated samples is due to the existence of pre-installed phosphates added by c-Abl. As noticed by us as well as other labs, CTD kinases don’t prefer to add phosphates to a hyper-phosphorylated CTD substrate. C-Abl adds phosphorylation heterogeneously along full-length CTD resulting in an apparent decrease in the number of phosphates incorporated by P-TEFb in the second reaction. Once CTD is fully phosphorylated with an average one phosphate per heptad, it is no longer a good substrate for P-TEFb. This is similar to what was observed in the Geyer Lab when all tyrosine residues are phosphorylated, P-TEFb no longer can add phosphate to the CTD (Czudnochowski, 2012). We also observed that when we phosphorylate the CTD with TFIIH until no phosphates are added (around 26 phosphates for yeast CTD), neither c-Abl nor P-TEFb can add additional phosphate to the substrate. The biological meaning of this trend requires further investigation but suggests the implication of temporal control in phosphorylation/dephosphorylation during eukaryotic transcription.

10) Can the authors quantify the overall levels of Ser2P and Ser5P when c-Abl-pretreated K7-CTD was used? Is Ser5P still the predominant form of products by p-TEFb? This would be an important parameter to conclude whether Y1P merely promotes Ser2P vs. Y1P alters Ser5P to Ser2P.

Yes, the tandem-treatment product species containing either pSer2 or pSer5 are quantified via the liquid chromatography trace in Figure 3G. Figure 3G peaks 7 and 8 contain the pSer2 species and are of higher intensity than peaks 6 and 9, the pSer5 containing species. PSer5 is no longer the predominant product formed by P-TEFb when pTyr1 marks are present. This is dramatically different from Figure 3E, where P-TEFb treatment alone generates pSer5 as its major product (Figure 3E peak 5). In these LC spectra, the product peptides are chemically similar to having the same base amino acid sequence and identical numbers of phosphates in species plotted in the same trace. Therefore, peak heights are comparable and reflect species abundance within the specific trace and can be used to derive relative abundance. When pTyr1 is present, pSer2 is the dominant product of P-TEFb phosphorylation.

11) Does p-TEFb cause further gel-shift of c-Abl-pretreated CTD?

Yes. We conducted an EMSA assay as suggested. The treatment of P-TEFb causes further gel shift for the CTD pre-treated with c-Abl. This is consistent with our MALDI-TOF results that show that up to 13 phosphates are added to the CTD with c-Abl treatment (Figure 2D) and further treatment with P-TEFb adds another ~13 phosphates (Figure 3A).

**Author response image 1. respfig1:** The native gel EMSA results for the tandem treatment of the CTD substrate treated by c-Abl alone or c-Abl followed by P-TEFb.

12) Descostes et al., 2014, showed that Tyr1P co-immunoprecipitated with Ser5P and Ser7P but not Ser2P or Thr4P. Can the authors reconcile this seeming discrepancy between the above observations and their main conclusion?

We agree with the result that pTyr1 occurs at the initiation stage of transcription, coincident with the pSer5 and pSer7. As seen in the pTyr1 ChIP-seq analysis, its existence is transient and precedes the apparent accumulation of phosphoryl-Ser2. During transcription, the appearance and disappearance of phosphorylated species are highly dynamic. It is difficult to capture the moment of transition from initiation to productive elongation. Instead, most of the species observed in currently published experiments indicate either initiation (Tyr1P, Ser5P, and Ser7P) or elongation/termination (Ser2P and Thr4P). Furthermore, the antibodies used in the co-immunoprecipitation are frequently interfered by flanking phosphorylation species and may further limit our ability to detect molecules at the initiation/elongation transition.

13) The authors should consider re-purifying the phosphorylated CTD after the c-Abl reaction so as to completely rule out the possibility that kinase-kinase interaction might cause the p-TEFb activity switch.

One major technical problem with pTyr1 CTD substrates and their proteolytic peptides is their propensity to fall out of solution. This is why such a large amount of material is needed for UVPD-MS analyses. Issues with material recovery are particularly difficult during affinity purification chromatography that would be required to re-purifying the phosphorylated CTD after the c-Abl reaction. In our experience, the combination of the affinity beads, phosphorylated substrate, coupled with the small reaction volumes feasible results in the pTyr1 containing substrate being almost entirely retained on the affinity beads. For these reasons, we utilized alternative approaches applying either a chemical inhibitor or heat inactivation. We are currently working with our mass spec collaborators to develop and optimize UVPD-MS method to reduce the requirement of the amount of starting material and hope to overcome that technical obstacle soon.

14) It wasn't clear to this reviewer why dmCTDs (WT and YtoF mutant) were used in part of Figure 1, then authors changed to yeast CTD for the rest of kinetics experiments. If similar results were obtained from both yeast and fly, it might be easier to show one complete set of data for consistency.

Thank you for pointing this out. The multiple banding in EMSA of YtoF mutants was first observed in *Drosophila melanogaster* CTD, which motivated us to do the ESI-MS and subsequent experiments investigating how the side-chain of Tyr1 affects other CTD kinases. Likely owing to the highly divergent nature of *Drosophila melanogaster* CTD, each phosphorylation species is well separated in EMSA. Because of this observation, we investigated if consensus CTD sequences also have altered phosphorylation pattern when Y is replaced by F using yeast CTD. The observation of multiple banding is indeed conserved but are much more difficult to detect and interpret in phosphorylated yeast CTD with the consistent consensus sequence. We realize that introducing *Drosophila melanogaster* CTD complicates the story. However, the reality is that if we had started with consensus yeast CTD with YtoF mutations, the presence of multiple bands might have been easily missed. After consideration, we decided to honestly describe the process and experiments that led us to our hypothesis about the influence of the chemical properties of residues at the Tyr1 position on CTD modifying enzymes. To clarify this motivation, we edited the manuscript to explain our thought process and the importance of the divergence of dmCTD to our data and detection of multiple species in EMSA.

15) The author should consider stressing that continuous Y1P inhibits p-TEFb-mediated phosphorylation (Czudnochowski, 2012) whereas c-Abl-pretreated CTD displays an alternate Y1P pattern which promotes Ser2P phosphorylation.

This is a great suggestion. We added this to the Discussion.